# NOISE PROMPT LEARNING: LEARNING THE WINNING TICKETS FOR DIFFUSION SAMPLING

## ABSTRACT

Text-to-image diffusion model is a popular paradigm that synthesizes personalized images by providing a text prompt and a random Gaussian noise. While people observe that some noises are winning tickets that can achieve better text-image alignment and higher human preference than others, we still lack a machine learning framework to obtain those winning noises. To learn winning noises for diffusion sampling, we mainly make three contributions in this paper. First, we identify a new concept termed the *noise prompt*, which aims at turning a random Gaussian noise into a winning noise ticket by adding a small desirable perturbation derived from the text prompt. Following the concept, we first formulate the *noise prompt learning* framework that systematically learns "prompted" winning noise tickets associated with a text prompt for diffusion models. Second, we design a noise prompt data collection pipeline and collect a large-scale *noise prompt dataset* (NPD) that contains 100k pairs of random noises and winning noises with the associated text prompts. With the prepared NPD as the training dataset, we trained a small *noise prompt network* (NPNet) that can directly learn to transform a random noise ticket into a winning noise ticket. The learned winning noise perturbation can be considered as a kind of prompt for noise, as it is rich in semantic information and tailored to the given text prompt. Third, our extensive experiments demonstrate the impressive effectiveness and generalization of NPNet on improving the quality of synthesized images across various diffusion models, including SDXL, DreamShaper-xl-v2-turbo, and Hunyuan-DiT. Moreover, NPNet is a small and efficient controller that acts as a plug-and-play module with very limited additional inference and computational costs, as it just provides a winning noise instead of a random noise without accessing the original pipeline.

## 1 INTRODUCTION

Image synthesis that are precisely aligned with given text prompts remains a significant challenge for text-to-image (T2I) diffusion models (Betker et al.; Chen et al., 2023; Rombach et al., 2022b; Peebles & Xie, 2023; Pernias et al., 2023). Previous studies (Yu et al., 2024; Wu et al., 2022; Toker et al., 2024; Kolors, 2024) have investigated the influence of text embeddings on the synthesized images and leveraged these embeddings for training-free image synthesis. It is well known that text prompts significantly matter to the quality and fidelity of the synthesized images. However, image synthesis is induced by both the text prompts and the noise. Variations in the noise can lead to substantial changes in the synthesized images, as even minor alterations in the noise input can dramatically influence the output (Xu et al., 2024; Qi et al., 2024). This sensitivity underscores the critical role that noise plays in shaping the final visual representation, affecting both the overall aesthetics and the semantic faithfulness between the synthesized images and the provided text prompt.

Recent studies (Lugmayr et al., 2022; Guo et al., 2024; Chen et al., 2024; Qi et al., 2024; Chefer et al., 2023) observe that some selected or optimized noises are winning tickets that can help the T2I diffusion models to produce images of better semantic faithfulness with text prompts, and can also improve the overall quality of the synthesized images. These methods (Chefer et al., 2023; Guo et al., 2024) incorporate extra modules like attention to reduce the truncate errors during the sampling process, showing promising results on the compositional generalization task. However, they are often not widely adopted in practice for several reasons. First, they often struggle to generally

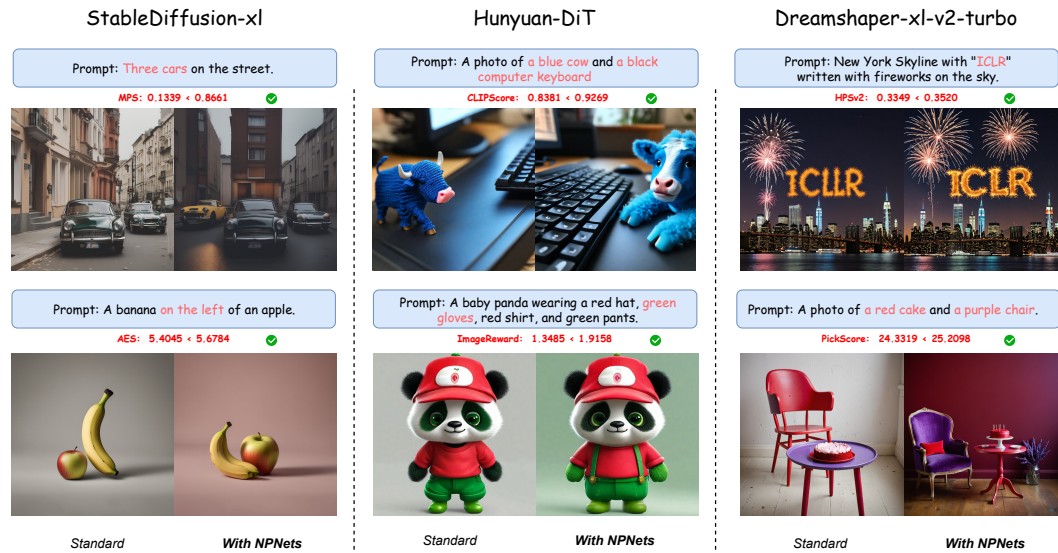

Figure 1: We visualize images synthesized by 3 different diffusion models and evaluate them using 6 human preference metrics. Images for each prompt are synthesized using the same random seed. These images with NPNet demonstrate a noticeable improvement in overall quality, aesthetic style, and semantic faithfulness, along with numerical improvements across all six metrics. More importantly, our NPNet is applicable to various diffusion models, showcasing strong generalization performance with broad application potential. More visualization results are in Appendix 14.

transfer to various benchmark datasets or diffusion models but only work for some specific tasks. Second, these methods often introduce significant time delays in order to optimize the noises during the reverse process. Third, they require in-depth modifications to the original pipelines when applied to different T2I diffusion models with varying architectures. Fourth, they need specific subject tokens for each prompt to calculate the loss of certain areas, which are unrealistic requirements for real users. These not only significantly complicate the original inference pipeline but also raise concerns regarding the generalization ability across various T2I diffusion models and datasets.

In light of the aforementioned research, we pose several critical questions: *1) Can we formulate obtaining the winning noise tickets as a machine learning problem so that we can predict them efficiently with only one model forward inference? 2) Can such a machine learning framework generalize well to various noises, prompts, and even diffusion models?* Fortunately, the answers are affirmative. In this paper, we mainly make three contributions:

First, we identify a new concept termed *noise prompt*, which aims at turning a random noise into a winning noise ticket by adding a small desirable perturbation derived from the text prompt. The winning noise perturbation can be considered as a kind of prompt for noise, as it is rich in semantic information and tailored to the given text prompt. Building upon this concept, we first formulate a *noise prompt learning* framework that systematically learns "prompted" winning noise tickets associated with text prompts for diffusion models.

Second, to implement the formulated *noise prompt learning* framework, we propose the training dataset, namely the *noise prompt dataset* (NPD), and the learning model, namely the *noise prompt network* (NPNet). Specifically, we design a noise prompt data collection pipeline via *re-denoise sampling*, a way to produce noise pairs. We also incorporate AI-driven feedback mechanisms to ensure that the noise pairs are highly valuable. This pipeline enables us to collect a large-scale training dataset for noise prompt learning, so the trained NPNet can directly transform a random Gaussian noise into a winning noise ticket to boost the performance of the T2I diffusion model.

Third, we conduct extensive experiments across various mainstream diffusion models, including StableDiffusion-xl (SDXL) (Podell et al., 2023), DreamShaper-xl-v2-turbo and Hunyuan-DiT (Li et al., 2024), with 7 different samplers on 3 different datasets. We evaluate our model by utilizing 6 human preference metrics including PickScore (Kirstain et al., 2023), Human Preference

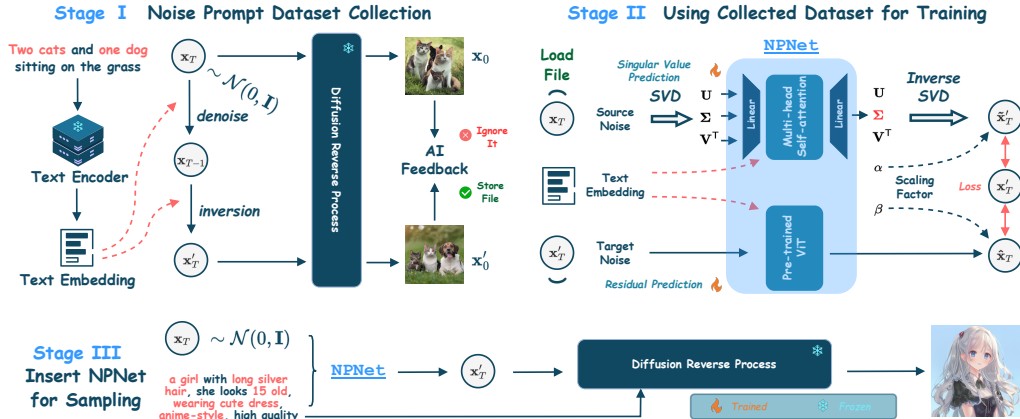

Figure 2: Our workflow diagram consists of three main stages. *Stage I:* We begin by denoising the original random Gaussian noise $\mathbf{x}_T$ to obtain $\mathbf{x}_{T-1}$, and then use DDIM-Inversion$(\cdot)$ to obtain inverse $\mathbf{x}'_T$ with more semantic information. Both synthesized images $\mathbf{x}_0$ and $\mathbf{x}'_0$ are filtered by the human preference model, such as HPSv2, to ensure the dataset is both diverse and representative. *Stage II:* After collecting NPD, we input the original noise (source noise) $\mathbf{x}_T$, inverse noise (target noise) $\mathbf{x}'_T$ and text prompt $\mathbf{c}$ into the NPNet, where the noises are processed by the *singular value predictor* and the *residual predictor*, and text prompt $\mathbf{c}$ is encoded by the text encoder $\mathcal{E}(\cdot)$ of the T2I diffusion model, resulting in the winning noise ticket. *Stage III:* Once trained, our NPNet can directly convert the random Gaussian noise into a winning noise ticket before inputting T2I diffusion models, boosting the performance of these models.

Score v2 (HPSv2) (Wu et al., 2023), Aesthetic Score (AES), ImageReward (Xu et al., 2023), CLIP-Score (Hessel et al., 2022) and Multi-dimensional Preference Score (MPS) (Zhang et al., 2024). As illustrated in Fig. 1, by leveraging the learned winning noise tickets, not only is the overall quality and aesthetic style of the synthesized images visually enhanced, but all metrics also show significant improvements, demonstrating the effectiveness and generalization ability of our NPNet. Furthermore, the NPNet is a compact and efficient neural network that functions as a plug-and-play module, introducing only a 3% increase in inference time per image compared to the standard pipeline, while requiring approximately 3% of the memory required by the standard pipeline. This efficiency underscores the practical applicability of NPNet in real-world scenarios.

## 2 PRELIMINARIES

We first present preliminaries about DDIM and DDIM Inversion and the classifier-free guidance. Due to the space constraints, we introduce the related work in Appendix B.

Given the Gaussian noise $\epsilon_t \sim \mathcal{N}(0, \mathbf{I})$, we denote the forward process of diffusion models as $\mathbf{x}_t = \alpha_t \mathbf{x}_0 + \sigma_t \epsilon_t$, where the $t \in \{0, 1, \cdots, T\}$. Here, $\alpha_t$ and $\sigma_t$ are predefined noise schedules, and $x_0$ is the original image.

**DDIM and DDIM Inversion.** Denoising diffusion implicit model (DDIM) (Song et al., 2023) is an advanced deterministic sampling technique, deriving an implicit non-Markov sampling process of the diffusion model. It allows for faster synthesis while maintaining the quality of synthesized samples. Its reverse process can be formulated as:

$$\mathbf{x}_{t-1} = \text{DDIM}(\mathbf{x}_t) = \alpha_{t-1}\left(\frac{\mathbf{x}_t - \sigma_t \epsilon_\theta(\mathbf{x}_t, t)}{\alpha_t}\right) + \sigma_{t-1}\epsilon_\theta(\mathbf{x}_t, t) \tag{1}$$

Using DDIM to add noise instead of applying Eq. 1 is called DDIM-Inversion:

$$\mathbf{x}_t = \text{DDIM-Inversion}(\mathbf{x}_{t-1}) = \frac{\alpha_t}{\alpha_{t-1}}\mathbf{x}_{t-1} + \left(\sigma_t - \frac{\alpha_t}{\alpha_{t-1}}\sigma_{t-1}\right)\epsilon_\theta(\mathbf{x}_t, t) \tag{2}$$

**Classifier-free Guidance.** Classifier-free guidance (CFG) (Ho & Salimans, 2021) allows for better control over the synthesis process by guiding the diffusion model towards desired conditions, such

as text prompt, to enhance the quality and diversity of synthesized samples. The predicted noise $\epsilon_{pred}$ with CFG at timestep $t$ can be formulated as:

$$\epsilon_{pred} = (\omega + 1)\epsilon_\theta(\mathbf{x}_t, t|\mathbf{c}) - \omega\epsilon_\theta(\mathbf{x}_t, t|\varnothing), \tag{3}$$

where we denote the $\mathbf{c}$ as the text prompt, $\omega$ as the CFG scale. Based on this, the denoised image $\mathbf{x}_{t-1}$ by using DDIM($\cdot$) can be written as:

$$\mathbf{x}_{t-1} = \alpha_{t-1}\left(\frac{\mathbf{x}_t - \sigma_t\left[(\omega + 1)\epsilon_\theta(\mathbf{x}_t, t|\mathbf{c}) - \omega\epsilon_\theta(\mathbf{x}_t, t|\varnothing)\right]}{\alpha_t}\right) + \sigma_{t-1}\left[(\omega + 1)\epsilon_\theta(\mathbf{x}_t, t|\mathbf{c}) - \omega\epsilon_\theta(\mathbf{x}_t, t|\varnothing)\right] \tag{4}$$

## 3 NOISE PROMPT LEARNING

In this section, we present the methodology of noise prompt learning, including NPD collection, NPNet design and training, as well as sampling with NPNet.

*Noise prompt* can be considered as a kind of special prompt, which aims at turning a random noise into a winning noise ticket by adding a small desirable perturbation derived from the text prompt. Analogous to text prompts, appropriate noise prompts can enable diffusion models to synthesize higher-quality images that are rich in semantic information. As illustrated on the left in Fig. 3, *text prompt learning* in large language models (Liu et al., 2021) focuses on learning how to transform a text prompt into a more desirable version. Similarly, *noise prompt learning* in our work seeks to learn how to convert the random Gaussian noise into the winning noise by adding a small, desirable perturbation derived from the text prompt. Using the winning noise, the diffusion model can synthesize images with higher quality and semantic faithfulness. Defining it as a machine learning problem, we are the first to formulate the *noise prompt learning* framework, as illustrated on the right in Fig. 3. Given the training set $\mathcal{D} := \left\{\mathbf{x}_{T_i}, \mathbf{x}'_{T_i}, \mathbf{c}_i\right\}_{i=1}^{|\mathcal{D}|}$ consisting of source

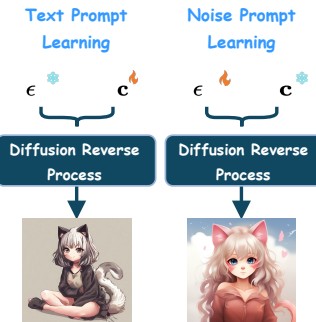

Figure 3: Paradigms of *text prompt learning* and *noise prompt learning*.

noises $\mathcal{X}$, target noises $\mathcal{X}'$ and text prompts $\mathcal{C}$, loss function $\ell$ and the neural network $\phi$, the general formula for the *noise prompt learning* task is:

$$\phi^* = \arg\min_\phi \mathbb{E}_{(\mathbf{x}_{T_i}, \mathbf{x}'_{T_i}, \mathbf{c}_i)\sim\mathcal{D}}[\ell(\phi(\mathbf{x}_{T_i}, \mathbf{c}_i), \mathbf{x}'_{T_i})]. \tag{5}$$

In summary, our goal is to learn the optimal neural network model $\phi^*$ trained on the training set $\mathcal{D}$. We present the data-training-inference workflow diagram with three stages in Fig. 3 and provide the pseudocodes for each stage in Appendix Alg. 1, Alg. 2 and Alg. 3, respectively.

### 3.1 NOISE PROMPT DATASET COLLECTION

In this subsection, we outline the training data collection pipeline, which consists of collecting noise pairs and AI-feedback-based data selection, as shown in Fig. 2 *Stage I*.

**Re-denoise Sampling Produces Noise Pairs.** *How to collect winning noise tickets with desirable semantic information?* Meng et al. (2023) reported that adding the random noise at each timestep during the sampling process and then re-denoising, leads to a substantial improvement in the semantic faithfulness of the synthesized images. This motivated us to propose a simple and direct approach called RE-DENOISE SAMPLING. Instead of directly adding noise to each timestep during the reverse process, we propose to utilize DDIM-Inversion($\cdot$) to obtain the noise from the previous step. Specifically, the joint action of DDIM-Inversion and CFG can induce the initial noise to attach semantic information. We denote the CFG scale within DDIM($\cdot$) and DDIM-Inversion($\cdot$) as $\omega_l$ and $\omega_w$, respectively. It is sufficient to ensure that the initial noise can be purified stably and efficiently by $\mathbf{x}'_t$=DDIM-Inversion(DDIM($\mathbf{x}_t$)) with $\omega_l > \omega_w$. Utilizing this method, the synthesized image from $\mathbf{x}'_t$ is more semantic information contained with higher fidelity, compared with the synthesized image from $\mathbf{x}_t$. We call the inverse noise $\mathbf{x}'_t$ target noise, and the noise $\mathbf{x}_t$ source noise. The visualization results are shown in Fig. 4. The mechanism behind this method is that DDIM-Inversion($\cdot$) injects semantic information by leveraging the CFG scale inconsistency. We present **theoretical understanding** of this mechanism in Theorem E.1 with the proof.

**Data Selection with the Human Preference Model.** While employing *re-denoise sampling* can help us collect noises with enhanced semantic information, it also carries the risk of introducing extra noises, which may lead to synthesizing images that do not achieve the quality of the originals. To mitigate this issue, we utilize a human preference model for data selection. This model assesses the synthesized images based on human preferences, allowing us to retain those noise samples that meet our quality standards. The reservation probability for data selection can be formulated as $\mathbb{I}[s_0 + m < s_0']$, where $m$ is the filtering threshold, $\mathbb{I}[\cdot]$, $s_0$ and $s_0'$ are the indicator function, human preference scores of denoised images from $\mathbf{x}_0$ and $\mathbf{x}_0'$, respectively. If the noise samples meet this criterion, we consider them to be valuable noise pairs and proceed to collect them.

By implementing this filtering process, we aim to keep a balance between leveraging the benefits of *re-denoise sampling* and maintaining the integrity of the synthesized outputs. For the selection strategies, we introduce them in Appendix Sec. D.1.

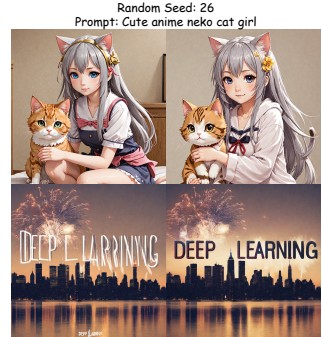

### 3.2 Noise Prompt Network

After data collection, we propose the architecture, training, inference of NPNet, as shown in Fig. 2 *Stage II* and *Stage III*.

**Architecture Design.** The architecture of NPNet consists of two key components, including *singular value prediction* and *residual prediction*, as shown in Fig. 2 *Stage II*.

The first key model component is *singular value prediction*. We obtain noise pairs through *re-denoise sampling*, a process that can be approximated as adding a small perturbation to the source noises. We observe that through the *singular value decomposition* (SVD), the singular vectors of $\mathbf{x}_T$ and $\mathbf{x}_T'$ exhibit remarkable similarity, albeit possibly in opposite directions, shown in Fig. 5, which may be partly explained by Davis-Kahan Theorem (Stewart, 1990; Xie et al., 2023). Building upon this observation, we design an architecture to predict the singular values of the target noise, illustrated in Fig. 2 Stage II. We denote $\phi(\cdot, \cdot, \cdot)$ as a ternary function that represents the sum of three inputs, $f(\cdot)$ as the linear layer function, and $g(\cdot)$ as the multi-head self-attention layer. The paradigm can be formulated as:

Figure 4: Visualization results about *re-denoise sampling*. *Re-denoise sampling* can help to inject semantic information of the text prompt into the original Gaussian noise, boosting the fidelity of synthesized images.

$$\mathbf{x}_T = U \times \Sigma \times V^\mathsf{T}, \quad \tilde{\mathbf{x}}_T = \phi(U, \Sigma, V^\mathsf{T}), \quad \tilde{\Sigma} = f(g(\tilde{\mathbf{x}}_T)), \quad \tilde{\mathbf{x}}_T' = U \times \tilde{\Sigma} \times V^\mathsf{T}, \quad (6)$$

where we denote $\tilde{\mathbf{x}}_T'$ as the predicted target noise. This model utilizes SVD inverse transformation to effectively reconstruct the target noise. By leveraging the similarities in the singular vectors, our model enhances the precision of the target noise restoration process.

The second key model component is residual prediction. In addition to *singular value prediction*, we also design an architecture to predict the residual between the source noise and the target noise, as illustrated in Fig. 2 Stage II. We denote $\varphi(\cdot)$ as the UpSample-DownConv operation, $\varphi'(\cdot)$ as the DownSample-UpConv operation, and the $\psi(\cdot)$ as the ViT model. The target noise incorporates semantic information from the text prompt $\mathbf{c}$ introduced through CFG. To facilitate the learning process, we inject this semantic information using the frozen text encoder $\mathcal{E}(\cdot)$ of the T2I diffusion model. This approach allows the model to effectively leverage the semantic information provided by the text prompt, ultimately improving the accuracy of the noise residual prediction. The procedure can be described as follows:

$$\mathbf{e} = \sigma(\mathbf{x}_T, \mathcal{E}(\mathbf{c})) \quad \hat{\mathbf{x}}_T = \varphi'(\psi(\varphi(\mathbf{x}_T + \mathbf{e})), \quad (7)$$

where we denote $\sigma(\cdot, \cdot)$ as AdaGroupNorm to ensure stability during the training process, and $\mathbf{e}$ as the normalized text embedding.

**Using Collected Dataset for Training.** The training procedure is also illustrated in Fig. 2 *Stage II*. To yield optimal results with two model components, we formulate the training loss as

$$\mathcal{L}_{\text{MSE}} = \text{MSE}(\mathbf{x}_T', \mathbf{x}_{T_{pred}}'), \text{ where } \mathbf{x}_{T_{pred}}' = \tilde{\mathbf{x}}_T' + \beta\hat{\mathbf{x}}_T, \quad (8)$$

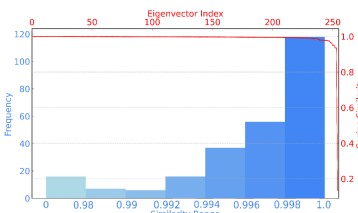 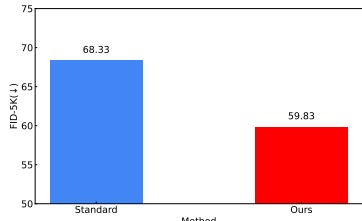

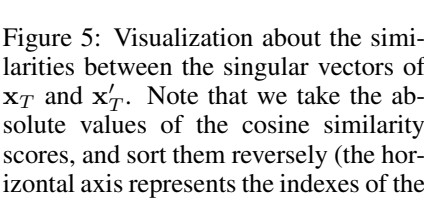

Figure 5: Visualization about the similarities between the singular vectors of $\mathbf{x}_T$ and $\mathbf{x}'_T$. Note that we take the absolute values of the cosine similarity scores, and sort them reversely (the horizontal axis represents the indexes of the singular vectors).

Figure 6: The FID comparison with 5000 images in class-conditional ImageNet with the resolution $512 \times 512$. The results validate the effectiveness of our NPNet on improving the conventional image quality metric.

$\beta$ is a trainable parameter used to adaptively adjust the weights of the predicted residuals, and $L$ as the MSE$(\cdot)$ loss function. For the nuanced adjustment in how much semantic information contributes to the model's predictions, $\mathbf{x}'_{T_{pred}} = \tilde{\mathbf{x}}'_T + \beta\hat{\mathbf{x}}_T$ can be rewritten as:

$$\mathbf{x}'_{T_{pred}} = \alpha\mathbf{e} + \tilde{\mathbf{x}}'_T + \beta\hat{\mathbf{x}}_T, \tag{9}$$

where we denote $\alpha$ as a trainable parameter. The values of these two parameters are shown in Appendix Table 13. we notice that $\alpha$ is very small, but it still plays a role in adjusting the influence of injected semantic information (the experimental results are shwon in Appendix Table 11). These two parameters, $\alpha$ and $\beta$, facilitate a refined adjustment of how much semantic information influences the model's predictions, enabling the semantic relevance between the text prompt and synthesized images, and the diversity of synthesized images.

**Insert NPNet for Sampling.** The inference procedure is illustrated in Fig. 2 *Stage III*. Once trained, our NPNet can be directly applied to the T2I diffusion model by inputting the initial noise $\mathbf{x_T}$ and prompt embedding $\mathbf{c}$ encoded by the frozen text encoder of the diffusion model. Our NPNet can effectively transform the original initial noise into the winning noise ticket. We also provide the example code on SDXL, shown in the Appendix Fig. 13.

## 4 EMPIRICAL ANALYSIS

In this section, we empirically study the effectiveness, the generalization, and the efficiency of our NPNet. We conduct a lot of experiments across various datasets on various T2I diffusion models, including SDXL, DreamShaper-xl-v2-turbo, and Hunyuan-DiT. Due to space constraints, we leave implementation details and additional experiments in Appendix A and D, respectively.

**Description of Training and Test Data.** We collect our NPD on Pick-a-pic dataset (Kirstain et al., 2023), which contains 1M prompts in its training set. We randomly choose 100k prompts as our training set. For each prompt, we randomly assign a seed value in $[0, 1024]$. For testing, we use three datasets, including the first 100 prompts from the Pick-a-Pic web application, the first 100 prompts from HPD v2 test set (Wu et al., 2023), and all 200 prompts from DrawBench (Saharia et al., 2022). For more details about the test data, please see in Appendix Fig. 18. We construct three training datasets collected from Pick-a-Pic, with 100k noise pairs, 80k noise pairs, and 600 noise pairs for SDXL, DreamShaper-xl-v2-turbo, and Hunyuan-DiT.

### 4.1 MAIN RESULTS

We evaluate our NPNet in three different T2I diffusion models. The main results[1] are shown in Table 1 and Table 2. In Table 1, we first evaluate our NPNet with SDXL and DreamShaper-xl-v2-turbo. The results demonstrate the impressive performance of our NPNet, almost achieving the best

---

[1]Method "Inversion" means *re-denoise sampling*.

Table 1: Main experiments on SDXL and DreamShaper-xl-v2-turbo over various datasets. Note that MPS calculates the preference scores between two images. We choose the standard sampling as the baseline and the MPS socre is always $0.5$, marked as "$-$".

| Model | Dataset | Method | PickScore | HPSv2 | AES | ImageReward | CLIPScore | MPS |
|---|---|---|---|---|---|---|---|---|
| SDXL | Pick-a-Pic | Standard | 21.6977 | 0.2848 | 6.0373 | 0.5801 | 0.8204 | - |
| | | Inversion | 21.7146 | 0.2857 | 6.0503 | 0.6327 | 0.8250 | 0.5141 |
| | | Repaint | 21.7799 | 0.2863 | 5.9875 | 0.6494 | 0.8327 | 0.5079 |
| | | NPNet (ours) | 21.8642 | 0.2868 | 6.0540 | 0.6501 | 0.8408 | 0.5214 |
| | DrawBench | Standard | 22.3118 | 0.2672 | 5.5952 | 0.6221 | 0.8077 | - |
| | | Inversion | 22.3751 | 0.2691 | 5.6017 | 0.6709 | 0.8081 | 0.5198 |
| | | Repaint | 22.3002 | 0.2696 | 5.5104 | 0.6407 | 0.8106 | 0.5204 |
| | | NPNet (ours) | 22.3828 | 0.2714 | 5.6034 | 0.7067 | 0.8153 | 0.5370 |
| | HPD | Standard | 22.8885 | 0.2971 | 5.9985 | 0.9663 | 0.8734 | - |
| | | Inversion | 22.8976 | 0.2978 | 5.9948 | 0.9739 | 0.8708 | 0.5303 |
| | | Repaint | 22.9116 | 0.2978 | 5.9948 | 0.9739 | 0.8775 | 0.5463 |
| | | NPNet (ours) | 22.9348 | 0.2988 | 5.9922 | 0.9881 | 0.8813 | 0.5602 |
| DreamShaper-xl-v2-turbo | Pick-a-Pic | Standard | 22.4168 | 0.3212 | 6.0161 | 0.9809 | 0.8267 | - |
| | | Inversion | 22.4000 | 0.3203 | 6.0236 | 1.0097 | 0.8277 | 0.4914 |
| | | NPNet (ours) | 22.7255 | 0.3269 | 6.0646 | 1.0674 | 0.8958 | 0.5234 |
| | DrawBench | Standard | 22.9803 | 0.3039 | 5.6735 | 0.9884 | 0.8186 | - |
| | | Inversion | 22.9467 | 0.3010 | 5.6852 | 0.9674 | 0.8189 | 0.4662 |
| | | NPNet (ours) | 23.1089 | 0.3078 | 5.7005 | 1.0814 | 0.8224 | 0.5353 |
| | HPD | Standard | 23.6858 | 0.3096 | 6.1408 | 1.2989 | 0.8868 | - |
| | | Inversion | 23.6731 | 0.3100 | 6.0811 | 1.3180 | 0.8912 | 0.4694 |
| | | NPNet (ours) | 23.6934 | 0.3408 | 6.1283 | 1.3598 | 0.8942 | 0.5249 |

Table 2: The fine-tuned NPNet showing strong cross-model generalization to enhancing Hunyuan-DiT, requiring only Hunyuan-DiT-produced 600 noise pairs for fine-tuning.

| Dataset | Method | PickScore | HPSv2 | AES | ImageReward | CLIPScore | MPS |
|---|---|---|---|---|---|---|---|
| Pick-a-Pic | Standard | 21.8205 | 0.2982 | 6.285 | 0.9133 | 0.8037 | - |
| | Inversion | 21.7684 | 0.2964 | 6.2756 | 0.8877 | 0.8021 | 0.4908 |
| | NPNet (ours) | 21.8368 | 0.2994 | 6.3470 | 1.0082 | 0.8101 | 0.5160 |
| DrawBench | Standard | 22.4457 | 0.2875 | 5.7152 | 0.9130 | 0.7940 | - |
| | Inversion | 22.4440 | 0.2875 | 5.7522 | 0.9286 | 0.7955 | 0.4925 |
| | NPNet (ours) | 22.4873 | 0.2889 | 5.8234 | 0.9620 | 0.8075 | 0.5193 |
| HPD | Standard | 22.8983 | 0.3087 | 6.0793 | 0.9922 | 0.8568 | - |
| | Inversion | 22.9052 | 0.3056 | 6.0802 | 1.0021 | 0.8617 | 0.5163 |
| | NPNet (ours) | 22.8880 | 0.3120 | 6.1573 | 1.0829 | 0.8694 | 0.5287 |



Figure 7: The winning rate comparison on SDXL across 3 datasets, including Pick-a-Pic, DrawBench and HPD v2 (HPD). The results reveal that our NPNet is the only one that can consistently transform random Gaussian noise into winning noise tickets, thereby enhancing the quality of the synthesized images, across nearly all evaluated datasets and metrics.

results across all 6 metrics and 3 datasets. We also evaluate our fine-tuned NPNet with Hunyuan-DiT, as shown in Table 2. For Hunyuan-DiT, we directly utilized the NPNet model trained on SDXL-produced NPD and fine-tune it on the Hunyuan-DiT-produced 600 noise pairs. Fine-tuned with only 600 samples, it can still achieve the highest results over the baselines on Hunyuan-DiT. This highlights the strong cross-model generalizability of our NPNet.

We also show the winning rate (the ratio of winning noise in the test set) on SDXL with our NPNet, shown in Fig. 7. We present more winning rate experiments of DreamShaper-xl-v2-turbo and Hunyuan-DiT in Appendix Fig. 10 and Appendix Fig. 15. These results again support that our NPNet is highly effective in transforming random Gaussian noise into winning noise tickets. In order to validate the effectiveness of our NPNet on improving the conventional image quality metric, we also calculate the FID (Heusel et al., 2018) of 5000 images on class conditional ImageNet with resolution $512 \times 512$ (please see more implementation details in Appendix D), shown in Fig. 6. We present more visualization results in Appendix Fig. 17.

Table 3: We conducted combinatorial experiments with other mainstream methods that can improve the alignment between the text prompts and synthesized images. The results indicate that our approach is orthogonal to these methods, allowing for joint usage to achieve improved performance.

| Methods | PickScore (↑) | HPSv2 (↑) | AES (↑) | ImageReward (↑) |
|---|---|---|---|---|
| Standard | 21.6977 | 0.2848 | 6.0373 | 0.5801 |
| DPO | 22.2180 | 0.3039 | 6.0159 | 0.8456 |
| DPO+NPNet (ours) | 22.2184 | 0.3056 | 6.0121 | 0.9403 |
| Standard | 21.0788 | 0.2538 | 5.9058 | 0.3404 |
| AYS | 21.5349 | 0.2724 | 6.0310 | 0.5074 |
| AYS+NPNet (ours) | 21.7578 | 0.2814 | 6.1239 | 0.5950 |

Table 4: Ablation studies of the proposed methods on SDXL on Pick-a-Pic dataset.

| Method | PickScore (↑) | HPSv2 (↑) | AES (↑) | ImageReward (↑) |
|---|---|---|---|---|
| Standard | 21.6977 | 0.2848 | 6.0373 | 0.5801 |
| NPNet *w/o singular value prediction* | 21.4972 | 0.2776 | 6.0164 | 0.4903 |
| NPNet *w/o residual prediction* | 21.8376 | 0.2855 | 6.0315 | 0.6305 |
| NPNet *w/o data selection* | 21.7319 | 0.2846 | 6.0375 | 0.6291 |
| NPNet | 21.8642 | 0.2868 | 6.0540 | 0.6501 |

## 4.2 ANALYSIS OF GENERALIZATION AND ROBUSTNESS

To validate the superior generalization ability of our NPNet, we conduct multiple experiments, covering experiments of cross-dataset, cross-model architecture, various inference steps, various random seed ranges, stochastic/deterministic samplers, and the integration of other methods.

**Generalization to Various Datasets and Models.** Although we trained our NPNets exclusively with the NPDs collected from the Pick-a-Pic dataset, the experimental results presented in Tables 1 and 2, and Fig. 7 demonstrate that our model exhibits strong cross-dataset generalization capabilities, achieving impressive results on other datasets as well. In addition to the fine-tuning experiments detailed in Table 2, we also applied the NPNet trained on NPD collected from SDXL to DreamShaper-xl-v2-turbo, evaluating the performance with our NPNet without any fine-tuning. The experiment results are shown in Appendix Table 9. These results indicate promising performance with our NPNet, underscoring the model's capability for cross-model generalization. We also conduct experiments on noise seed range in Appendix 14, the results demonstrate that our NPNet exhibits strong generalization capabilities across the out-of-distribution random seed ranges.

**Generalization to Stochastic/Deterministic Samplers.** When collecting NPD on SDXL, we use the deterministic DDIM sampler. However, whether the NPNet can effectively perform with stochastic samplers is crucial. To investigate our NPNet's performance across various sampling methods, we evaluated 7 different samplers, using NPNet trained on the NPD collected from SDXL, whose sampler is DDIM. The results shown in Fig. 8 and Appendix Table 5 suggest that our NPNet is adaptable and capable of maintaining high performance even when subjected to various levels of randomness in the sampling process, further validating the generalization of our NPNet.

**Orthogonal Experiment.** To explore whether our model can be combined with other approaches, which aim at enhancing the semantic faithfulness of synthesized images, such DPO (Rafailov et al., 2024) and AYS (Sabour et al., 2024), we conduct combination experiments, shown in Table 3. Note that AYS only releases the code under the inference step 10, so we conduct the combinatorial experiment with AYS under the inference step 10. The results indicate that our method is orthogonal to these works, allowing for joint usage to further improve the quality of synthesized images.

**Robustness to the Hyper-parameters.** We study how the performance of NPNet is robust to the hyper-parameters. We first evaluate the performance of our NPNet under various inference steps, as illustrated in Fig. 9, Appendix Fig. 10, Appendix Fig. 11, and Appendix Fig. 15. These results highlight the generalization and versatility of our NPNet is robust to the common range of inference steps. inference steps. Such consistency suggests that the model is well-tuned to adapt to different conditions, making it effective for a wide range of applications. We also do exploration studies on the other hyper-parameters, such as batch size, the training epochs, and CFG values in Appendix

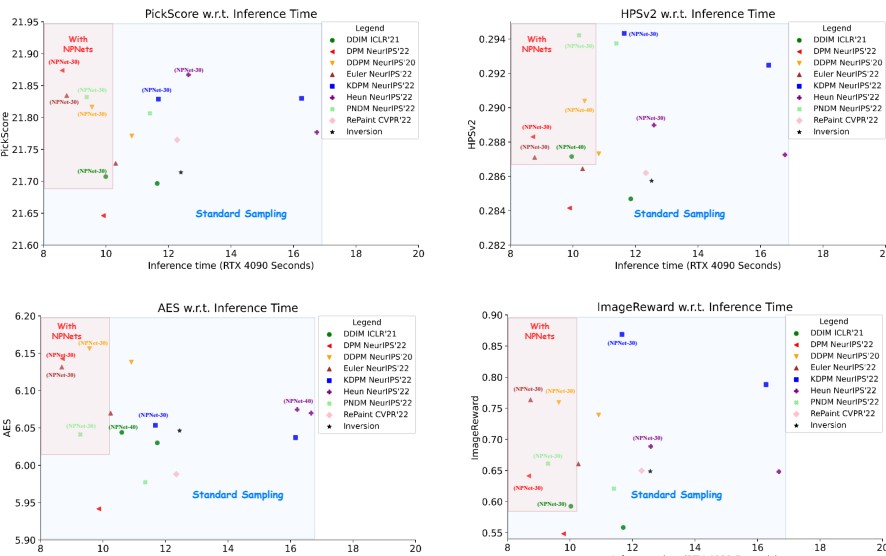

Figure 8: We evaluate our NPNet with 7 samplers on SDXL in Pick-a-Pic dataset, including both the deterministic sampler and stochastic sampler (with a default inference step 50). "NPNet-30" means the inference step is 30 with NPNet. The red area in the top left corner of the image represents the results of efficient high-performance methods, while the experimental results of NPNet are nearly in that same region. It highlights that NPNet is capable of synthesizing higher-quality images with fewer steps and consuming less time. Moreover, the results demonstrate the generalization ability of our NPNet across different samplers.

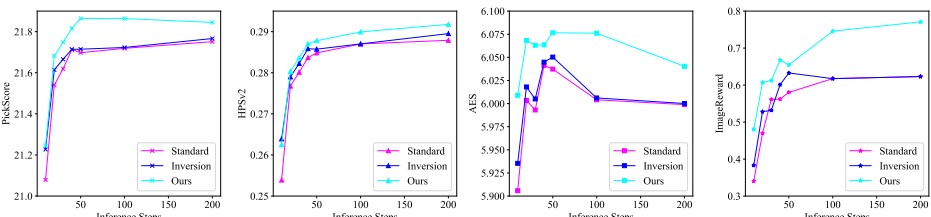

Figure 9: Visualization of performance w.r.t. inference steps on SDXL on Pick-a-Pic dataset. With our NPNet, T2I diffusion models can have superior performance under various inference steps.

Table 10. The studied optimal settings are the batch size with 64, the training epochs with 30, and the CFG value with 5.5. Moreover, we explore the influences of different amounts of training samples, shown in Appendix Table 12.

**Ablation Studies.** We conduct ablation studies about the architecture designs of NPNet in Table 4. The results show that both *singular value prediction* and *residual prediction* contribute to the final optimal results, while the *singular value prediction* component plays a more important role. We also empirically verify the effectiveness of data selection strategies in Appendix Tables 7 and 8.

## 4.3 EFFICIENCY ANALYSIS AND DOWNSTREAM TASK EXPLORATION

**Efficiency Analysis.** Given the plug-and-play nature of NPNet, it is essential to discuss the memory consumption and inference latency. Remarkably, our model achieves significant performance improvements even with fewer inference steps and reduced time costs in Fig. 8. Even when operating at the same number of inference steps in Table 5, our model introduces only a 0.4-second delay while synthesizing high-quality images, demonstrating its efficiency. Additionally, Appendix Fig. 16 shows its memory consumption is mere 500 MB, highlighting its resource-friendly design. Our model not only delivers superior results but also exhibit significant application potential and practical value due to the impressive deployment efficiency.

Table 5: Experiments on different samplers *w.r.t.* inference time cost on SDXL. The NPNet trained on noise samples produced by the deterministic sampler DDIM, demonstrates impressive generalization to non-deterministic samplers, incurring only minimal additional time costs.

| Methods | | PickScore (↑) | HPSv2 (↑) | AES (↑) | ImageReward (↑) | Time Cost(second per image) |
|---|---|---|---|---|---|---|
| DDIMScheduler (Song et al., 2023) | Standard | 21.6977 | 0.2848 | 6.0373 | 0.5801 | 11.69 |
| | NPNet (ours) | 21.8642 | 0.2868 | 6.054 | 0.6501 | 12.10 |
| DPMSolverMultistepScheduler (Lu et al., 2022) | Standard | 21.6598 | 0.2841 | 5.9513 | 0.5501 | 9.84 |
| | NPNet (ours) | 21.7171 | 0.2881 | 5.9744 | 0.6730 | 10.43 |
| DDPMScheduler (Ho et al., 2020) | Standard | 21.7851 | 0.2872 | 6.1353 | 0.7291 | 10.86 |
| | NPNet (ours) | 21.9040 | 0.2924 | 6.1505 | 0.7850 | 11.43 |
| EulerAncestralDiscreteScheduler (Karras et al., 2022) | Standard | 21.7263 | 0.2866 | 6.0740 | 0.6701 | 10.28 |
| | NPNet (ours) | 21.8353 | 0.2896 | 6.0886 | 0.8505 | 10.86 |
| PNDMScheduler (Karras et al., 2022) | Standard | 21.7830 | 0.2935 | 5.9809 | 0.6281 | 11.40 |
| | NPNet (ours) | 21.8054 | 0.2974 | 6.0256 | 0.6758 | 11.82 |
| KDPM2AncestralDiscreteScheduler (Karras et al., 2022) | Standard | 21.8174 | 0.2922 | 6.0382 | 0.7759 | 16.25 |
| | NPNet (ours) | 21.9303 | 0.2962 | 6.0951 | 0.8478 | 16.73 |
| HeunDiscreteScheduler (Karras et al., 2022) | Standard | 21.8316 | 0.2871 | 6.0705 | 0.6433 | 16.74 |
| | NPNet (ours) | 21.8499 | 0.2898 | 6.0892 | 0.7331 | 17.04 |

**Exploration of Downstream Task.** We explored the potential of integrating NPNet with downstream tasks, specifically by combining it with ControlNet (Zhang et al., 2023) for controlled image synthesis. As a plug-and-play module, our NPNet can be seamlessly incorporated into ControlNet. Visualization results in Appendix Fig. 12 demonstrate that this integration leads to the synthesis of more detailed and higher-quality images, highlighting the effectiveness of our approach.

## 5 DISCUSSION

**Limitations.** Although our experimental results have demonstrated the superiority of our method, the limitations still exist. As a machine learning framework, our method also faces classic challenges from training data quality and model architecture design. First, noise prompt data quality sets the performance limit of our method. The data quality is heavily constrained by *re-denoise sampling* and data selection, but lack comprehensive understanding. For example, there exists the potential risk that the proposed data collection pipeline could introduce extra bias due to the AI-feedback-based selection. Second, the design of NPNet is still somewhat rudimentary. While ablation studies support each component of NPNet, it is highly possible that more elegant and efficient architectures may exist and work well for the novel noise prompt learning task. Optimizing model architectures for this task still lacks principled understanding and remain to be a challenge.

**Future Directions.** Our work has various interesting future directions. First, it will be highly interesting to investigate improved data collection methods in terms of both performance and trustworthiness. Second, we will design more streamlined structures rather than relying on a parallel approach with higher performance or higher efficiency. For example, we may directly utilize a pretrained diffusion model to synthesize winning noise more precisely. Third, we will further analyze and improve the generalization of our method, particularly in the presence of out-of-distribution prompts or even beyond the scope of T2I tasks.

## 6 CONCLUSION

In this paper, we introduce a novel concept termed the *noise prompt*, which aims to transform random Gaussian noise into a winning ticket noise by incorporating a small perturbation derived from the text prompt. Building upon this, we firstly formulate a *noise prompt learning* framework that systematically obtains "prompted" winning tickets associated with a text prompt for diffusion models, by constructing a *noise prompt dataset* collection pipeline that incorporates HPSv2 to filter our data and designing several backbones for our *noise prompt models*. Our extensive experiments demonstrate the superiority of NPNet, which is plug-and-play, straightforward, and nearly time-efficient, while delivering significant performance improvements. This model possesses considerable application potential and practical significance. We believe that the future application scope of NPNet will be broad and impactful, encompassing video, 3D content, and seamless deployment of real AIGC business products, thereby making a meaningful contribution to the community.

**Ethics Statement.** We propose NPNet, a lightweight neural network designed to enhance the semantic faithfulness of images generated by various diffusion models, necessitate careful consideration of several ethical issues. Although NPNet does not directly involve human subjects, we are committed to ensuring that its applications respect user autonomy and promote positive outcomes. We emphasize transparency in our dataset releases, ensuring that all datasets used are ethically sourced and compliant with applicable laws, while actively working to mitigate potential biases inherent in the data. We also prioritize the privacy and security of any data utilized, adhering to data protection regulations to safeguard user information. Given NPNet's significant commercial potential, we strive to apply this technology responsibly, ensuring that its applications yield positive societal benefits.

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

## A    IMPLEMENTATION DETAILS

In this section, we present the benchmarks, evaluation metrics and the relevant contents we used in the main paper to facilitate a comprehensive understanding of our model's performance. This overview will help contextualize our results and provide clarity on how we assessed the effectiveness of our approach.

### A.1    BENCHMARKS

In our main paper, we conduct experiments across three popular text-to-image datasets.

**Pick-a-Pic.**    Pick-a-Pic (Kirstain et al., 2023) is an open dataset designed to collect user preferences for images synthesized from text prompts. The dataset is gathered through a user-friendly web application that allows users to synthesize images and select their preferences. Each data sample includes a text prompt, two synthesized images, and a label indicating which the user prefers or a tie if there is no clear preference. The Pick-a-Pic dataset contains over 500,000 examples covering 35,000 unique prompts. Its advantage lies in the fact that the data comes from real users, reflecting their genuine preferences rather than relying on paid crowd workers

**DrawBench.**    DrawBench is a newly introduced benchmark dataset designed for in-depth evaluation of text-to-image synthesis models. It contains 200 carefully crafted prompts categorized into 11 groups, testing the models' abilities across various semantic attributes, including compositionality, quantity, spatial relationships, and handling complex text prompts. The design of DrawBench allows for a multidimensional assessment of model performance, helping researchers identify strengths and weaknesses in image synthesis. By comparing with other models, DrawBench provides a comprehensive evaluation tool for the text-to-image synthesis field, facilitating a deeper understanding of synthesis quality and image-text alignment.

**HPD v2.**    The Human Preference Dataset v2 (Wu et al., 2023) is a large-scale, cleanly annotated dataset focused on user preferences for images synthesized from text prompts. It contains 798,090 binary preference choices involving 433,760 pairs of images, aiming to address the limitations of existing evaluation metrics that fail to accurately reflect human preferences. HPD v2 eliminates potential biases and provides a more comprehensive evaluation capability, with data sourced from multiple text-to-image synthesis models and real images.

For testing, we use these three datasets, including the first 100 prompts subset from the Pick-a-Pic web application, 100 prompts from HPD v2 test set, and 200 prompts from DrawBench. The detailed information of the test sets is shown in Fig. 18.

### A.2    EVALUATION METRICS

In our main paper, we mainly include 6 evaluation metrics to validate the effectiveness of our NPNet.

**PickScore.**    PickScore is a CLIP-based scoring function trained from the Pick-a-Pic dataset, which collects user preferences for synthesized images. It achieves superhuman performance when predicting user preferences. PickScore aligns well with human judgments, and together with Pick-a-Pic's natural distribution prompts, enables much more relevant text-to-image model evaluation than evaluation standards, such as FID (Heusel et al., 2018) over MS-COCO (Lin et al., 2015).

**HPSv2.**    Human Preference Score v2 (HPSv2) is an advanced preference prediction model by fine-tuning CLIP (Radford et al., 2021) on Human Preference Dataset v2 (HPD v2). This model aims to align text-to-image synthesis with human preferences by predicting the likelihood of a synthesized image being preferred by users, making it a reliable tool for evaluating the performance of text-to-image models across diverse image distribution.

**AES.**    Aesthetic Score (AES) are derived from a model trained on the top of CLIP embeddings with several extra multilayer perceptron (MLP) layers to reflect the visual appeal of images. This

metric can be used to evaluate the aesthetic quality of synthesized images, providing insights into how well they align with human aesthetic preferences.

**ImageReward.** ImageReward (Xu et al., 2023) is a human preference reward model specifically designed for evaluating text-to-image synthesis. It is trained on a large dataset of human comparisons, allowing it to effectively encode human preferences. The model assesses synthesized images based on various criteria, including alignment with the text prompt and overall aesthetic quality. ImageReward has been shown to outperform traditional metrics like Inception Score (IS) (Barratt & Sharma, 2018) and Fréchet Inception Distance (FID) in correlating with human judgments, making it a promising automatic evaluation metric for text-to-image synthesis.

**CLIPScore.** CLIPScore (Hessel et al., 2022) leverages the capabilities of the CLIP model, which aligns images and text in a shared embedding space. By calculating the cosine similarity between the image and text embeddings, CLIPScore provides a mearsure of how well a synthesized image corresponds to its textual description. While CLIPScore is effective in assessing text-image alignment, it may not fully capture the nuances of human preferences, particularly in terms of aesthetic quality and detail.

**MPS.** Multi-dimensional Preference Score (MPS) (Zhang et al., 2024), the first multi-dimensional preference scoring model for the evaluation of text-to-image models. The MPS introduces the preference condition module upon CLIP model to learn these diverse preferences. It is trained based on the Multi-dimensional Human Preference (MHP) Dataset, which comprises 918,315 human preference choices across four dimensions, including aesthetics, semantic alignment, detail quality and overall assessment on 607,541 images, providing a more comprehensive evaluation of synthesized images. MPS calculates the preference scores between two images, and the sum of the two preference scores equals 1.

## A.3 T2I DIFFUSION MODELS

In the main paper, we totally use 3 T2I diffusion models, including StableDiffusion-xl (SDXL) (Podell et al., 2023), DreamShaper-xl-v2-turbo (DreamShaper), and Hunyuan-DiT (DiT) (Li et al., 2024).

**StableDiffusion-xl.** StableDiffusion-xl (SDXL) is an advanced generative model, building upon the original Stable Diffusion architecture. This model leverages a three times larger UNet backbone, and utilizes a refinement model, which is used to improve the visual fidelity of samples synthesized by SDXL using a post-hoc image-to-image technique. SDXL is designed to synthesize high-resolution images from text prompts, demonstrating significant improvements in detail, coherence, and the ability to represent complex scenes compared to its predecessors.

**DreamShaper-xl-v2-turbo.** DreamShaper-xl-v2-turbo, a fine-tuned version on SDXL, is a text-to-image model designed for high-quality image synthesis, focusing on faster inference time and enhanced image synthesis capabilities. DreamShaper-xl-v2-turbo maintains the high-quality image output characteristic of its predecessor, while its turbo enhancement allows for quicker synthesis cycles. The overall style of the synthesized images leans towards fantasy, while it achieves a high level of authenticity when realism is required.

**Hunyuan-DiT.** Hunyuan-DiT is a text-to-image diffusion transformer with fine-grained understanding of both English and Chinese. With careful design of the model architecture, it can perform multi-turn multimodal dialogue with users to synthesized high-fidelity images, under the refinement of the Multimodal Large Language Model.

## A.4 HYPER-PARAMETER SETTINGS

Our method is straightforward and intuitive, and the parameter settings for the entire experiment are also very simple, with epoch 30, and batch size 64 for all experiments. We conduct experiments on three T2I diffusion models, including SDXL, DreamShaper-xl-v2-turbo, and Hunyuan-DiT, with

CFG $\omega_l$ 5.5, 3.5, and 5.0 respectively. The inverse CFG $\omega_w$ is 1.0 for all three models. To collect training data, the inference steps are 10, 4, and 10 for SDXL with DDIM inverse scheduler, Dreamshaper-xl-v2-turbo with DPMSolver inverse scheduler, and Hunyuan-DiT with DDIM inverse scheduler, respectively. The human preference model we use to filter the data is the HPSv2, and the filtering threshold $k$ equals 0. Unless otherwise specified, all quantitative experiments and synthesized images in this paper are conducted and synthesize with inference steps 50, respectively. All experiments are conducted using $1\times$ RTX 4090 GPUs, and all these noise pairs are collected with inference step 10 to construct NPDs.

## B  RELATED WORK

Synthesizing images that are precisely aligned with given text prompts remains a significant challenge for Text-to-Image (T2I) diffusion models. To deal with this problem, several works explore training-free improvement strategies, by optimizing the noises during the diffusion reverse process.

Lugmayr et al. (2022) utilizes a pre-trained unconditional diffusion model as a generative prior and alters the reverse diffusion iterations based on the unmasked regions of the input image. Meng et al. (2023) observe that denoising the noise with inversion steps can generate better images compared with the original denoising process. Based on that, Qi et al. (2024) aims to reduce the truncate errors during the denoising process, by increase the cosine similarity between the initial noise and the inversed noise in an end-to-end way. It introduces significant time costs, and the synthesized images may be over-rendered, making it difficult to use in practical scenarios.

Another research direction introduces extra modules to help optimize the noises during the reverse process. Chefer et al. (2023) introduce the concept of Generative Semantic Nursing (GSN), and slightly shifts the noisy image at each timestep of the denoising process, where the semantic information from the text prompt is better considered. InitNO (Guo et al., 2024) consists of the initial latent space partioning and the noise optimization pipeline, responsible for defining valid regions and steering noise navigation, respectively. Such methods are not universally applicable, we discuss this in Appendix C

Table 6: Comparison results with InitNO on StableDiffusion-v1-4 (Rombach et al., 2022a) on Pick-a-Pic dataset. We directly apply NPNet trained for SDXL, and remove the embedding e to StableDiffusion-v1-4.

|  | PickScore (↑) | HPSv2 (↑) | AES (↑) | ImageReward (↑) |
|---|---|---|---|---|
| Standard | 19.1732 | 0.1949 | 5.4575 | -1.2273 |
| InitNO | 16.5039 | 0.1447 | 5.3116 | -2.0566 |
| NPNet (ours) | 19.2474 | 0.1954 | 5.5151 | -0.9589 |

Unlike previous approaches, we are the first to reframe this task as a learning problem. we directly learn to prompt the initial noise into the winning ticket noise to address this issue, by training a universal Noise prompt network (NPNet) with our noise prompt dataset (NPD). Our NPNet operates as a plug-and-play module, with very limited memory cost and negligible inference time cost, produce images with higher preference scores and better alignment with the input text prompts effectively.

## C  DISCUSSION WITH THE PREVIOUS WORKS

We previously mentioned that these methods (Chefer et al., 2023; Guo et al., 2024) optimize the noise during the reverse process by incorporating additional modules. These methods have shown promising results in tasks involving compositional generalization. However, these methods often struggle to transfer to other datasets and models, making them not universally applicable. These approaches require the manual interest subject tokens, necessitating extensive test to identify the optimal tokens for a given sentence, which complicates their application across different datasets. Furthermore, modifying the model pipeline usually requires in-depth code changes, making it difficult to achieve straightforward plug-and-play integration with other models. Moreover, these methods demand multiple rounds of noise optimization during the reverse process, resulting in significant time consumption.

In contrast, our approach addresses these challenges from multiple perspectives, offering a more flexible and universal solution. It is capable of cross-model and cross-dataset applications, provides plug-and-play functionality, and incurs minimal time overhead. The experimental results are shown in Table 6. We follow the code in Guo et al. (2024), and manually provide the subject tokens following Chefer et al. (2023). We conduct the experiments on StableDiffusion-v1-4 (Rombach et al., 2022a) on Pick-a-Pic dataset. We directly apply the NPNet trained for SDXL to StableDiffusion-v1-4. The results demonstrate the superiority of our NPNet.

# D ADDITIONAL EXPERIMENT RESULTS

## D.1 EXPLORATION OF DATA SELECTION STRATEGIES

Since the target noise collected through *re-denoise sampling* is not always of high quality, it is crucial to choose an appropriate method for data filtering. Effective selection ensures that only high-quality noise pairs are used, which is essential for training the NPNet, affecting the model's performance and reliability. For this reason, we conduct experiments on the choice of human preference model to filter our data, shown in Table 7, here the filtering threshold $m = 0$. The results demonstrate that using HPSv2 ensures data diversity, allowing the filtered data to enhance the model's performance effectively. This approach helps maintain a rich variety of training samples, which contributes to the model's generalization ability and overall effectiveness.

Table 7: To collect valuable samples, we explore the data selection with different human preference models on SDXL with inference steps 10 on Pick-a-Pic dataset."Standard*" here means no human preference model is applied.

| Method | Filter Rate | PickScore (↑) | HPSv2 (↑) | AES (↑) | ImageReward (↑) |
|---|---|---|---|---|---|
| Standard* | - | 21.2184 | 0.2595 | 5.9608 | 0.4047 |
| PickScore | 34.28% | 21.2301 | 0.2590 | 5.9750 | 0.3256 |
| HPSv2 | 23.88% | 21.2409 | 0.2601 | 5.9675 | 0.4247 |
| AES | 47.62% | 21.2195 | 0.2583 | 5.9636 | 0.4323 |
| ImageReward | 35.02% | 21.2139 | 0.2570 | 5.958 | 0.3239 |
| PickScore+HPSv2 | 41.93% | 21.2298 | 0.2575 | 5.9514 | 0.3859 |
| PickScore+ImageReward | 52.57% | 21.1413 | 0.2597 | 5.9936 | 0.4219 |
| All | 74.41% | 21.2100 | 0.2595 | 5.9608 | 0.4047 |

We also explore the influence under difference filtering thresholds $m$, the results are shown in Table 8. Our findings reveal that while increasing the filtering threshold $m$ can improve the quality of the training data, it also results in the exclusion of a substantial amount of data, ultimately diminishing the synthesizing diversity of the final NPNet.

Table 8: Experiments on the HPSv2 filtering threshold. We conducted experiments on SDXL on Pick-a-Pic dataset to investigate the impact of adding a threshold during the filtering process, like $s_0 + m < s'_0$, where $s_0$ and $s'_0$ are the human preference scores of denoising images $\mathbf{x}_0$ and $\mathbf{x}'_0$.

| Threshold | Filter Rate | PickScore (↑) | HPSv2 (↑) | AES (↑) | ImageReward (↑) |
|---|---|---|---|---|---|
| $m = 0$ | 23.88% | 21.8642 | 0.2868 | 6.0540 | 0.6621 |
| $m = 0.005$ | 41.21% | 21.7861 | 0.2879 | 6.0703 | 0.6030 |
| $m = 0.01$ | 50.19% | 21.7219 | 0.2864 | 6.0766 | 0.6160 |
| $m = 0.02$ | 83.47% | 21.8227 | 0.2878 | 6.0447 | 0.6546 |

## D.2 EVALUATE THE QUALITY OF SYNTHESIZED IMAGES

To validate the quality of the synthesized images with our NPNet, we calculate the FID[2] of 5000 images in class-conditional ImageNet with the resolution $512 \times 512$ on SDXL, shown in Fig. 6. Note

---

[2]We follow the code in https://github.com/GaParmar/clean-fid

that we just synthesize the "fish" class in the ImageNet dataset, whose directory ids are [n01440764, n01443537, n01484850, n01491361, n01494475, n01496331, n01498041]. The main fish class contains sub-class labels, including "tench", "Tinca tinca", "goldfish", "Carassius auratus", "great white shark", "white shark", "man-eater", "man-eating shark", "Carcharodon carcharias", "tiger shark", "Galeocerdo cuvieri", "hammerhead", "hammerhead shark", "electric ray", "crampfish", "numb-fish", "torpedo", "stingray". Each time, we randomly choose one prompt with postfix "a class in ImageNet", in order to synthesize ImageNet-like images. The results reveal that with our NPNet, the T2I diffusion models can synthesize images with higher quality than the standard ones.

### D.3 GENERALIZATION AND ROBUSTNESS

In this subsection, we provide more experiments to validate the generalization ability and robustness of our NPNet.

Table 9: Generalization on different diffusion models. We train our NPNet with NPD collected from SDXL. We apply it directly to DreamShaper-xl-v2-turbo on Pick-a-Pic dataset. Our results show promising performance, highlighting the model's capability for cross-model generalization.

| Inference Steps | | PickScore (↑) | HPSv2 (↑) | AES (↑) | ImageReward (↑) |
|---|---|---|---|---|---|
| 4 | Standard | 21.5790 | 0.2902 | 5.9172 | 0.5312 |
| | NPNet (ours) | 21.6129 | 0.2920 | 5.9159 | 0.5846 |
| 10 | Standard | 22.3961 | 0.3216 | 6.0296 | 0.9667 |
| | NPNet (ours) | 22.4054 | 0.3224 | 6.0320 | 0.9792 |
| 30 | Standard | 22.4235 | 0.3233 | 6.0116 | 0.9897 |
| | NPNet (ours) | 22.4386 | 0.3248 | 6.0054 | 1.0018 |
| 50 | Standard | 22.4168 | 0.3212 | 6.0161 | 0.9809 |
| | NPNet (ours) | 22.4623 | 0.3225 | 6.0033 | 0.9986 |

**Generalization to Models, Datasets and Inference Steps.**    In Table 9, we directly apply the NPNet for SDXL to DreamShaper-xl-v2-turbo without fine-tuning on the corresponding data samples. Even so, our NPNet achieves nearly the best performance across arbitrary inference steps, demonstrating the strong generalization capability of our model. Besides, we also present the winning rate of DreamShaper-xl-v2-turbo and Hunyuan-DiT across 3 different datasets, as presented in Appendix Fig. 10. These experimental results indicate that our method has a high success rate in transforming random Gaussian noise into winning noise, highlighting the effectiveness of our approach.

**Generalization to Random Seeds.**    As we mentioned in Sec. 3.2, the random seed range for the training set is $[0, 1024]$, while the random seed range for the test set is $[0, 100]$. This discrepancy may lead to our NPNet potentially overfitting on specific random seeds. To evaluate the performance of our NPNet under arbitrary random seeds, we artificially modified the seeds in the test set. The experimental results on the Pick-a-Pic dataset are presented in Table 14, demonstrating that our NPNet maintains strong performance across a variety of random seed conditions, making it suitable for diverse scenarios in real-world applications. The results demonstrate that our NPNet exhibits strong generalization capabilities across the out-of-distribution random seed ranges.

**Robustness to Inference Steps and Hyper-parameters.**    In Fig. 11, we conduct the experiments on DreamShaper-xl-v2-turbo and Hunyuan-DiT under various inference steps. The curve representing our method consistently remains at the top, demonstrating that our model achieves the best performance across various inference steps, further validating the robustness of our approach. To further support our claims, we present the winning rate of SDXL, DreamShaper-xl-v2-turbo and Hunyuan-DiT under various inference in two different datasets, shown in Fig. 15. These promising results validate the effectiveness of our NPNet.

**Robustness to Hyper-parameters.**    We also conduct the experiments on different hyper-parameter settings, including the CFG value, batch size and training epochs, shown in Table 10. It reveals that the optimal setting of these parameters are CFG 5.5, batch size 64, and training epoch 30. For all the

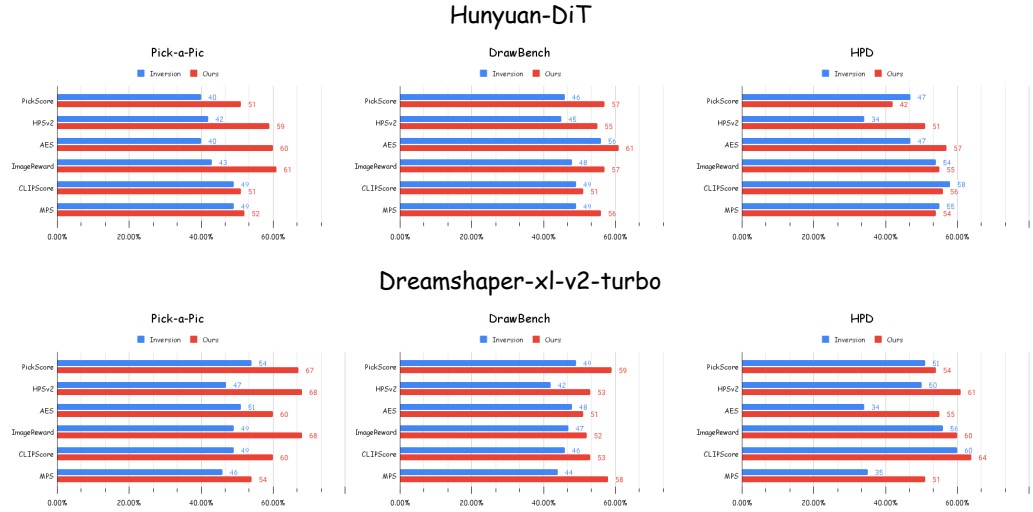

Figure 10: The winning rate comparison on DreamShaper-xl-v2-turbo and Hunyun-DiT across 3 datasets, including Pick-a-Pic, DrawBench and HPD v2 (HPD) with inference steps 50. The results demonstrate the superiority of our NPNet.

experiments in the paper, we all use this setting. Moveover, we explore the influence of the number of training samples, shown in Table 12, we believe that a large dataset can ensure data diversity and improve the model's robustness and generalization ability.

Table 10: Ablation studies of the hyper-parameters on SDXL on Pick-a-Pic dataset.

| Hyper-parameters | | PickScore (↑) | HPSv2 (↑) | AES (↑) | ImageReward (↑) |
|---|---|---|---|---|---|
| | 5 | 21.7731 | 0.2862 | 6.0688 | 0.6584 |
| Epochs | 10 | 21.7634 | 0.2867 | 6.0629 | 0.6059 |
| | 15 | 21.6927 | 0.2862 | 6.0721 | 0.5874 |
| | 30 | 21.8642 | 0.2868 | 6.054 | 0.6501 |
| | $\omega_1 = 1$ | 20.1131 | 0.2180 | 6.0601 | -0.5130 |
| Guidance Scale | $\omega_1 = 3$ | 21.5250 | 0.2728 | 6.0880 | 0.4449 |
| | $\omega_1 = 5.5$ | 21.8642 | 0.2868 | 6.0540 | 0.6501 |
| | $\omega_1 = 7$ | 21.8111 | 0.2912 | 6.0529 | 0.7031 |
| | $bs = 16$ | 21.7656 | 0.2874 | 6.0677 | 0.6080 |
| Batch Size | $bs = 32$ | 21.6876 | 0.2868 | 6.0483 | 0.6547 |
| | $bs = 64$ | 21.8642 | 0.2868 | 6.0540 | 0.6501 |

## D.4 EFFICIENCY ANALYSIS AND ABLATION STUDIES

**Efficiency Analysis.** As a plug-and-play module, it raises concerns about potential increases in inference latency and memory consumption, which can significantly impact its practical value. In addition to Fig. 8 presented in the main paper, we also measure the time required to synthesize each image under the same inference step conditions, shown in main paper Table 5. Our model achieves a significant improvement in image quality with only a 0.4-second inference delay. Additionally, as shown in Fig. 16, our model requires just 500 MB of extra memory. These factors highlight the lightweight and efficient nature of our model, underscoring its broad application potential.

**Ablation Studies.** We explore the influence of the text embedding term **e**. Although in Table 13, the value of $\alpha$ is very small, the results in Table 11 still demonstrate the importance of this term. It can facilitate a refined adjustment of how much semantic information influences the model's predictions, enabling the semantic relevance between the text prompt and synthesized images, and the diversity of the synthesized images.

# E    THEORETICAL SUPPORT OF RE-DENOISE SAMPLING

In main paper Sec. 3.1, we utilize *re-denoise sampling* to produce noise pairs. we propose to utilize DDIM-Inversion$(\cdot)$ to obtain the noise from the previous step. Specifically, the joint action of DDIM-Inversion and CFG can induce the initial noise to attach semantic information. The mechanism behind this method is that DDIM-Inversion$(\cdot)$ injects semantic information by leveraging the guidance scale in classifier-free guidance (CFG) inconsistency:

**Theorem E.1.** *Given the initial Gaussian noise* $\mathbf{x}_T \sim \mathcal{N}(0, \mathbf{I})$ *and the operators* DDIM-Inversion*(·) and* DDIM*(·). Using re-denoise sampling, we can obtain that:*

$$\mathbf{x}'_T = \mathbf{x}_T + \frac{\alpha_T \sigma_{T-k} - \alpha_{T-k}\sigma_T}{\alpha_{T-k}} \left[ (\omega_l - \omega_w)(\epsilon_\theta(\mathbf{x}_{T-\frac{k}{2}}, T-\frac{k}{2}|\mathbf{c}) - \epsilon_\theta(\mathbf{x}_{T-\frac{k}{2}}, T-\frac{k}{2}|\varnothing)) \right]. \quad (10)$$

*where* $k$ *stands for the DDIM sampling step,* $\mathbf{c}$ *is the text prompt, and* $\omega_l$ *and* $\omega_w$ *are CFG at the timestep* $T$ *and CFG at timestep* $T-k$, *respectively.*

*Proof.* One step *re-denoise sampling* represents one additional step forward sampling and one step reverse sampling against the initial Gaussian noise, which can be denoted as

$$\mathbf{x}'_T = \text{DDIM-Inversion}(\text{DDIM}(\mathbf{x}_T)), \quad (11)$$

where DDIM-Inversion$(\cdot)$ refers to the sampling algorithm in Eqn. 1 when $\mathbf{x}_t$ and $\mathbf{x}_{t-1}$ are interchanged. We can rewrite it in forms of linear transformation:

$$\mathbf{x}'_T = \alpha_T \left( \frac{\mathbf{x}_{T-k} - \sigma_{T-k}\epsilon_\theta(\mathbf{x}_{T-k}, T-k)}{\alpha_{T-k}} \right) + \sigma_T \epsilon_\theta(\mathbf{x}_{T-k}, T-k)$$

$$\mathbf{x}'_T = \alpha_T \left( \frac{\alpha_{T-k} \left( \frac{\mathbf{x}_T - \sigma_T \epsilon_\theta(\mathbf{x}_T, T)}{\alpha_T} \right) + \sigma_{T-k}\epsilon_\theta(\mathbf{x}_T, T) - \sigma_{T-k}\epsilon_\theta(\mathbf{x}_{T-k}, T-k)}{\alpha_{T-k}} \right) + \sigma_T \epsilon_\theta(\mathbf{x}_{T-k}, T-k)$$

$$\mathbf{x}'_T = \mathbf{x}'_T - \sigma_T \epsilon_\theta(\mathbf{x}_T, T) + \frac{\alpha_T \sigma_{T-k}}{\alpha_{T-k}}\epsilon_\theta(\mathbf{x}_T, T) - \frac{\alpha_T \sigma_{T-k}}{\alpha_{T-k}}\epsilon_\theta(\mathbf{x}_{T-k}, T-k) + \sigma_T \epsilon_\theta(\mathbf{x}_{T-k}, T-k)$$

$$\mathbf{x}'_T = \mathbf{x}'_T + \frac{\alpha_T \sigma_{T-k} - \alpha_{T-k}\sigma_T}{\alpha_{T-k}} \left[ \epsilon_\theta(\mathbf{x}_T, T) - \epsilon_\theta(\mathbf{x}_{T-k}, T-k) \right], \quad (12)$$

where $k$ stands for the DDIM sampling step. Substitute $\epsilon_\theta(\mathbf{x}_t, t) = (\omega+1)\epsilon_\theta(\mathbf{x}_t, t|\mathbf{c}) - \omega\epsilon_\theta(\mathbf{x}_t, t|\varnothing)$ into Eq. 12, we can obtain

$$\mathbf{x}'_T = \mathbf{x}'_T + \frac{\alpha_T \sigma_{T-k} - \alpha_{T-k}\sigma_T}{\alpha_{T-k}} \Big[ (\omega_l + 1)\epsilon_\theta(\mathbf{x}_T, T|\mathbf{c}) - \omega_l \epsilon_\theta(\mathbf{x}_T, T|\varnothing))$$

$$- (\omega_w + 1)\epsilon_\theta(\mathbf{x}_{T-k}, T-k|\mathbf{c}) + \omega_w \epsilon_\theta(\mathbf{x}_{T-k}, T-k|\varnothing)) \Big]. \quad (13)$$

Where $\omega_l$ and $\omega_w$ refer to the classifier-free guidance scale at the timestep $T$ and the classifier-free guidance scale at timestep $T - k$, respectively. $\mathbf{c}$ stands for the text prompt (*i.e.*, condition). Consider the first-order Taylor expansion $\epsilon_\theta(\mathbf{x}_{T-k}, T-k|\mathbf{c}) = \epsilon_\theta(\mathbf{x}_{T-\frac{k}{2}}, T-\frac{k}{2}|\mathbf{c}) + \frac{\mathbf{x}_{T-k} - \mathbf{x}_{T-\frac{k}{2}}}{2} \frac{\partial \epsilon_\theta(\mathbf{x}_{T-\frac{k}{2}}, T-\frac{k}{2}|\mathbf{c})}{\partial \mathbf{x}_{T-\frac{k}{2}}} + \frac{k}{2} \frac{\partial \epsilon_\theta(\mathbf{x}_{T-\frac{k}{2}}, T-\frac{k}{2}|\mathbf{c})}{\partial T-\frac{k}{2}} + \mathcal{O}(\left(\frac{k}{2}\right)^2)$ and $\epsilon_\theta(\mathbf{x}_T, T|\mathbf{c}) = \epsilon_\theta(\mathbf{x}_{T-\frac{k}{2}}, T-\frac{k}{2}|\mathbf{c}) + \frac{\mathbf{x}_T - \mathbf{x}_{T-\frac{k}{2}}}{2} \frac{\partial \epsilon_\theta(\mathbf{x}_{T-\frac{k}{2}}, T-\frac{k}{2}|\mathbf{c})}{\partial \mathbf{x}_{T-\frac{k}{2}}} - \frac{k}{2} \frac{\partial \epsilon_\theta(\mathbf{x}_{T-\frac{k}{2}}, T-\frac{k}{2}|\mathbf{c})}{\partial T-\frac{k}{2}} + \mathcal{O}(\left(\frac{k}{2}\right)^2)$, when $\mathbf{x}_T$ satisfies the condition $\left\| \frac{\|\|\mathbf{x}_T - \mathbf{x}_{T-k}\|\|}{k} \right\| \leq L$, where $L < +\infty$, Eq. 13 can be transformed into:

$$\mathbf{x}'_T = \mathbf{x}'_T + \frac{\alpha_T \sigma_{T-k} - \alpha_{T-k}\sigma_T}{\alpha_{T-k}} \left[ (\omega_l - \omega_w)(\epsilon_\theta(\mathbf{x}_{T-\frac{k}{2}}, T-\frac{k}{2}|\mathbf{c}) - \epsilon_\theta(\mathbf{x}_{T-\frac{k}{2}}, T-\frac{k}{2}|\varnothing)) \right]. \quad (14)$$

The proof is complete. □

By using Eqn. 14, when there is a gap between $\omega_l$ and $\omega_w$, *re-denoise sampline* can be considered as a technique to inject semantic information under the guidance of future timestep ($t = T - \frac{k}{2}$) CFG into the initial Gaussian noise.

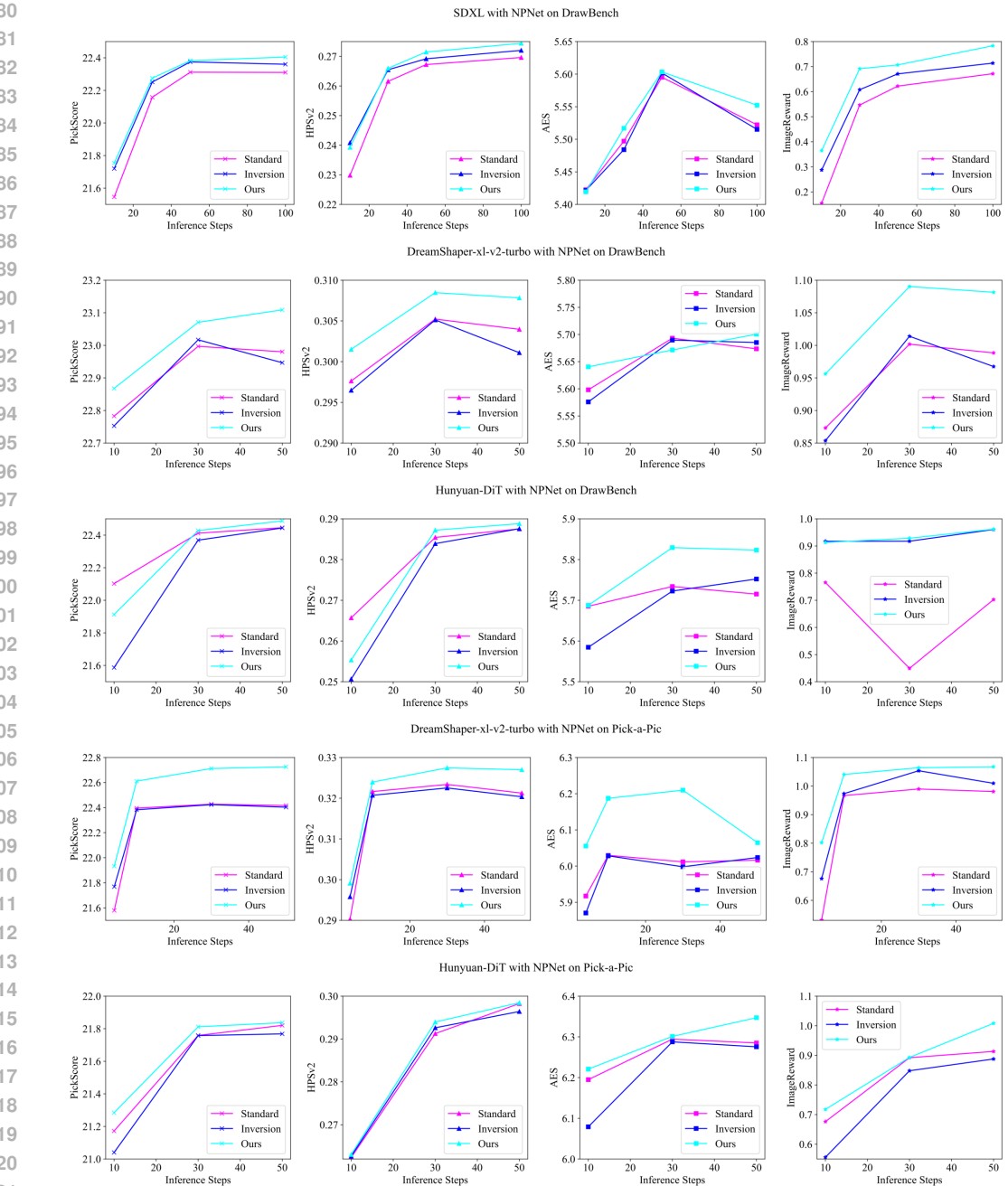

Figure 11: Visualization of performance w.r.t inference steps on SDXL, DreamShaper-xl-v2-turbo and Hunyuan-DiT on Pick-a-Pic dataset and DrawBench dataset. The results demonstrate the strong generalization ability of our NPNet.

Table 11: We explore the influence of text embedding $\mathbf{e}$. The results reveal that text embedding $\mathbf{e}$ is crucial in *noise prompt learning*, which aims to inject the semantic information into the noise.

| Method | PickScore (↑) | HPSv2 (↑) | AES (↑) | ImageReward (↑) |
|---|---|---|---|---|
| NPNet *w/o text embedding* $\mathbf{e}$ | 21.7244 | 0.2870 | 6.0513 | 0.6214 |
| NPNet | 21.8642 | 0.2868 | 6.0540 | 0.6501 |

Table 12: In order to explore the scaling law (Kaplan et al., 2020) in NPNet, we train our NPNet with different numbers of training samples on SDXL on Pick-a-Pic dataset.

| Numbers of Training Samples | Method | PickScore (↑) | HPSv2 (↑) | AES (↑) | ImageReward (↑) | CLIPScore (↑) | MPS (↑) |
|---|---|---|---|---|---|---|---|
| | Standard | 21.6977 | 0.2848 | 6.0373 | 0.5801 | 0.8204 | - |
| 3W | NPNet (ours) | 21.8187 | 0.2878 | 6.075 | 0.6726 | 0.8217 | 0.5063 |
| 6W | NPNet (ours) | 21.7457 | 0.2865 | 6.0392 | 0.6456 | 0.8198 | 0.516 |
| 10W | NPNet (ours) | 21.8642 | 0.2868 | 6.054 | 0.6501 | 0.8408 | 0.5215 |

Table 13: The values of the two trainable parameters $\alpha$ and $\beta$.

| Model | $\alpha$ | $\beta$ |
|---|---|---|
| SDXL | 1.00E-04 | -0.0189 |
| DreamShaper-xl-v2-turbo | 7.00E-05 | 0.0432 |
| Hunyuan-DiT | 2.00E-04 | 0.0018 |

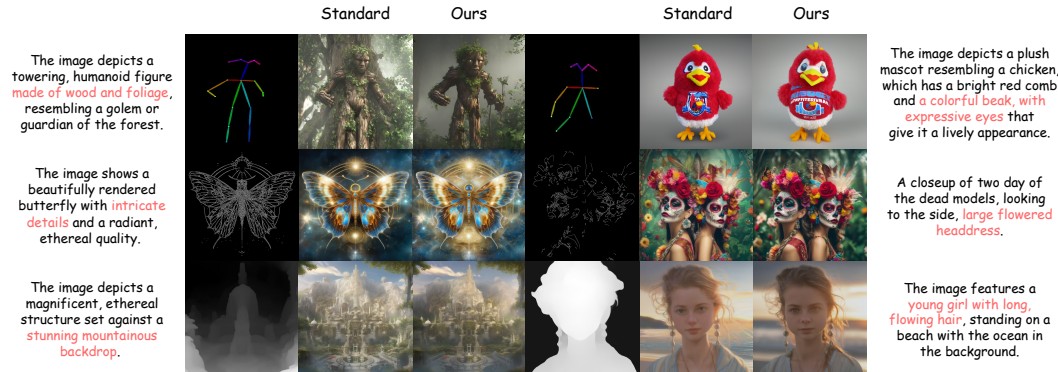

Figure 12: ControlNet visualization with our NPNet on SDXL, including conditions like openpose, canny and depth. *Middle* is the standard method, and *right* is our result. Our NPNet can be directly applied to the corresponding downstream tasks without requiring any modifications to the pipeline of the T2I diffusion model.

```
1
2   /* ---SDXL Code Example----*/
3
4   # initialize the pipeline , scheduler and NPNet
5   pipe = StableDiffusionXLPipeline.from_pretrained(model_id)
6   pipe.scheduler = DDIMScheduler.from_config(pipe.scheduler.config)
7   noise_model = NPNet()
8
9   # sample the initial noise
10  initial_noise =  torch.randn(latent_shape, dtype=dtype)
11
12  # get the wining noise ticket
13  prompt_embeds, _ = pipe.encode_prompt(prompt)
14  winning_noise = NPNet(prompt_embeds=prompt_embeds, initial_noise=initial_noise)
15
16  image =  pipe(prompt=prompt, height=height, width=width, guidance_scale=guidance_scale,
                 num_inference_steps=num_steps, latents=winning_noise).images[0]
```

Figure 13: Example inference code with NPNet on SDXL. Our NPNet operates as a plug-and-play module, which can be easily implemented.

---

**Algorithm 1:** Noise Prompt Dataset Collection

---

1: **Input:** Timestep $t \in [0, \cdots, T]$, random Gaussian noise $\mathbf{x}_T$, text prompt $\mathbf{c}$, DDIM operateor $\mathbf{DDIM}(\cdot)$, DDIM inversion operator $\mathbf{DDIM\text{-}Inversion}(\cdot)$, human preference model $\Phi$ and filtering threshold $m$.
2: **Output:** Source noise $\mathbf{x}_T$, target noise $\mathbf{x}'_T$ and text prompt $\mathbf{c}$.
3: Sample Gaussian noise $x_T$
4: *# re-denoise sampling, see Sec. 3.1 in the main paper*
5: $\mathbf{x}_{T-1} = \mathbf{DDIM}(\mathbf{x}_T)$
6: $\mathbf{x}'_T = \mathbf{DDIM\text{-}Inversion}(\mathbf{x}_{T-1})$
7: *# standard diffusion reverse process*
8: $\mathbf{x}_0 = \mathbf{DDIM}(\mathbf{x}_T)$
9: $\mathbf{x}'_0 = \mathbf{DDIM}(\mathbf{x}'_T)$
10: *# data filtering via the human preference model, see Sec. 3.1*
11: **if** $\Phi(\mathbf{x}_0, \mathbf{c}) + m < \Phi(\mathbf{x}'_0, \mathbf{c})$ **then**
12:     store $(\mathbf{x}_T, \mathbf{x}'_T, \mathbf{c})$
13: **end if**

---

**Algorithm 2:** Noise Prompt Network Training

---

1: **Input:** *Noise prompt dataset* $\mathcal{D} := \{\mathbf{x}_{T_i}, \mathbf{x}'_{T_i}, \mathbf{c}_i\}_{i=1}^{|\mathcal{D}|}$, *noise prompt model* $\phi$ parameterized by *singular value predictor* $f(\cdot)$ and *residual predictor* $g(\cdot, \cdot)$, the frozen pre-trained text encoder $\mathcal{E}(\cdot)$ from diffusion model, normalization layer $\sigma(\cdot, \cdot)$, MSE loss function $\ell$, and two trainable parameters $\alpha$ and $\beta$.
2: **Output:** The optimal *noise prompt model* $\phi^*$ trained on the training set $\mathcal{D}$.
3: *# singular value prediction*, see equation 6
    $\tilde{\mathbf{x}}'_{T_i} = f(\mathbf{x}_{T_i})$
4: *# residual prediction*, see equation 7
    $\hat{\mathbf{x}}_{T_i} = g(\mathbf{x}_{T_i}, \mathbf{c}_i)$
5: *# see equation 9*
    $\mathbf{x}'_{T_{pred_i}} = \alpha \sigma(\mathbf{x}_{T_i}, \mathcal{E}(\mathbf{c}_i)) + \tilde{\mathbf{x}}'_{T_i} + \beta \hat{\mathbf{x}}_{T_i}$
6: $\mathcal{L}_i = \ell(\mathbf{x}'_{T_{pred_i}}, \mathbf{x}'_{T_i})$
7: update $\phi$
8: **return** $\phi^*$

---

**Algorithm 3:** Inference with Noise Prompt Network

---

1: **Input:** Text prompt $\mathbf{c}$, the trained *noise prompt network* $\phi^*(\cdot, \cdot)$ and the diffusion model $f(\cdot, \cdot)$.
2: **Output:** The golden clean image $\mathbf{x}'_0$.
3: Sample Gaussian noise $x_T$
4: *# get the winning noise ticket*
    $\mathbf{x}'_{T_{pred}} = \phi^*(\mathbf{x}_T, \mathbf{c})$
5: *# standard inference pipeline*
    $\mathbf{x}'_0 = f(\mathbf{x}'_{T_{pred}}, \mathbf{c})$
6: **return** $\mathbf{x}'_0$

---

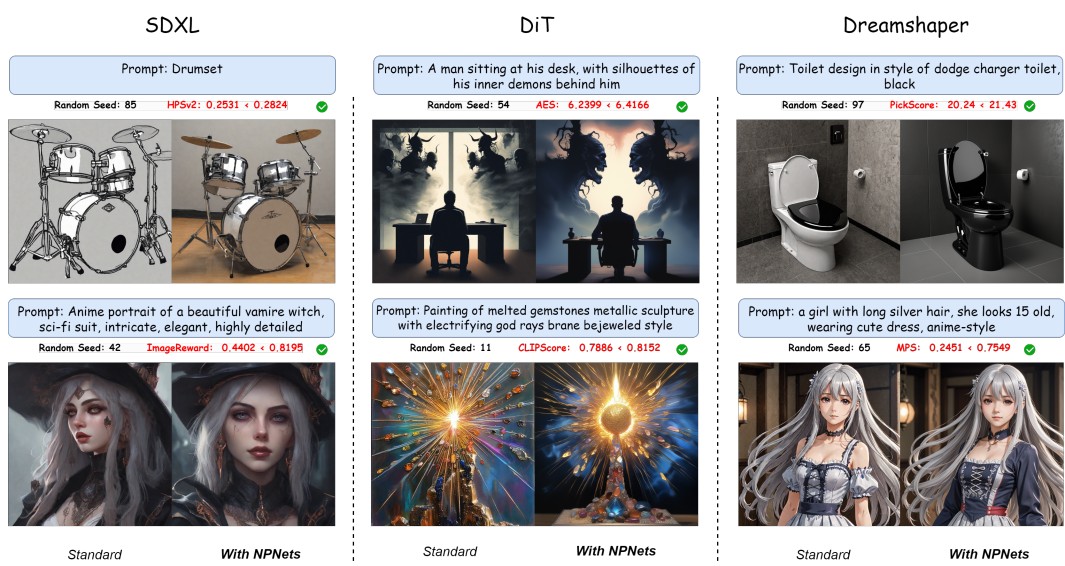

Figure 14: We visualized images synthesized by 3 different diffusion models and evaluated them using 6 human preference metrics. Images for each prompt are synthesized using the same random seed. These images with NPNet demonstrated a noticeable improvement in overall quality, aesthetic style, and semantic faithfulness, along with numerical improvements across all six metrics. More importantly, our NPNet is applicable to various diffusion models, showcasing strong generalization performance with broad application potential.

Table 14: Random seed generalization experiments on SDXL with difference inference steps on Pick-a-Pic dataset. The random seeds of our training set range from $[0, 1024]$, containing the random seeds of our test set. To explore the generalization ability of NPNet on out-of-distribution random seeds, we manually adjust the random seed range of the test set.

| Inference Steps | Random Seed Range | | PickScore (↑) | HPSv2 (↑) | AES (↑) | ImageReward (↑) |
|---|---|---|---|---|---|---|
| 50 | [0, 1024] (Original) | Standard | 21.6977 | 0.2848 | 6.0373 | 0.5801 |
| | | Inversion | 21.7146 | 0.2857 | 6.0503 | 0.6327 |
| | | NPNet (ours) | 21.8642 | 0.2868 | 6.0540 | 0.6501 |
| | [2500, 3524] | Standard | 21.6301 | 0.2857 | 5.9748 | 0.6709 |
| | | Inversion | 21.7071 | 0.2875 | 5.9875 | 0.7092 |
| | | NPNet (ours) | 21.8059 | 0.2902 | 5.9917 | 0.8083 |
| | [5000, 6024] | Standard | 21.7388 | 0.2882 | 6.0534 | 0.7802 |
| | | Inversion | 21.7815 | 0.2904 | 6.0418 | 0.7605 |
| | | NPNet (ours) | 21.8282 | 0.2909 | 6.4220 | 0.7984 |
| | [7500, 7524] | Standard | 21.7017 | 0.2912 | 6.0251 | 0.7871 |
| | | Inversion | 21.7811 | 0.2918 | 6.0541 | 0.8269 |
| | | NPNet (ours) | 21.8142 | 0.2902 | 6.0641 | 0.8953 |
| 100 | [0, 1024] (Original) | Standard | 21.7088 | 0.2870 | 6.0041 | 0.6176 |
| | | Inversion | 21.7230 | 0.2870 | 6.0061 | 0.6173 |
| | | NPNet (ours) | 21.8635 | 0.291 | 6.0761 | 0.7457 |
| | [2500, 3524] | Standard | 21.6951 | 0.2873 | 5.9946 | 0.6922 |
| | | Inversion | 21.7405 | 0.2892 | 5.9863 | 0.7186 |
| | | NPNet (ours) | 21.8089 | 0.2904 | 5.9977 | 0.7875 |
| | [5000, 6024] | Standard | 21.8077 | 0.2840 | 6.0489 | 0.7957 |
| | | Inversion | 21.8484 | 0.2866 | 6.0374 | 0.7994 |
| | | NPNet (ours) | 21.8796 | 0.2916 | 6.0576 | 0.8387 |
| | [7500, 7524] | Standard | 21.7346 | 0.2867 | 6.0347 | 0.7766 |
| | | Inversion | 21.8017 | 0.2875 | 6.0600 | 0.8233 |
| | | NPNet (ours) | 21.8592 | 0.2912 | 6.0502 | 0.8979 |

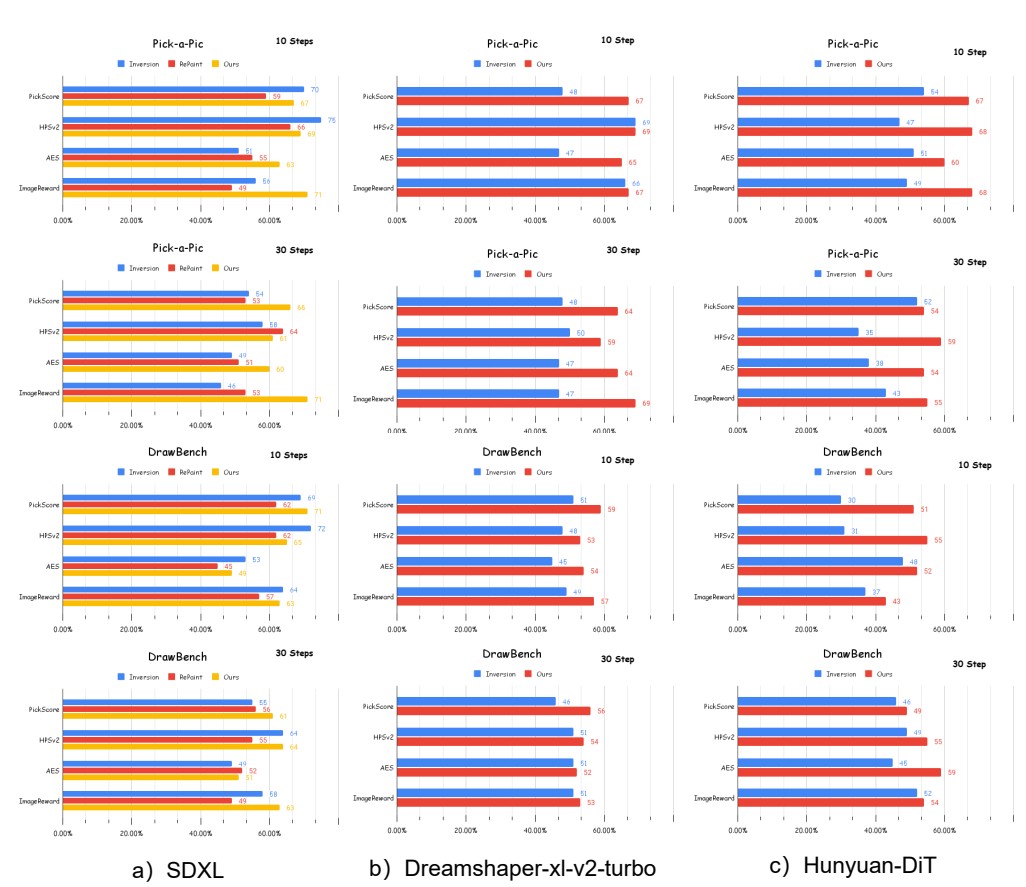

a) SDXL
b) Dreamshaper-xl-v2-turbo
c) Hunyuan-DiT

Figure 15: The winning rate comparison on SDXL, DreamShaper-xl-v2-turbo and Hunyuan-DiT across **2** datasets, including DrawBench and HPD v2 (HPD). The results reveal that our NPNet is more effective in transforming random Gaussian noise into winning noise tickets in different inference steps across different datasets.

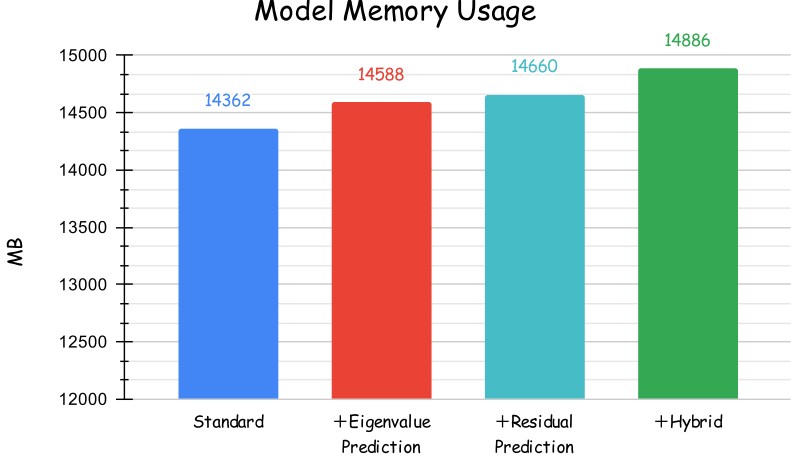

Figure 16: Our NPNet requires only about 500MB, illustrating the light-weight and efficiency of our model.

Two cats and one dog sitting on the grass.

Photo of an athlete cat explaining it's latest scandal at a press conference to journalists.

A donut underneath a toilet.

A type of digital currency in which a record of transactions is maintained...

Three cats and one dog sitting on the grass.

Standard     Inversion     RePaint     Ours

Figure 17: Visualization results about different baseline methods on SDXL.

```
0, A curious cat exploring a haunted mansion, 54
1, a spanish water dog breed as arthur morgan from red dead redemption, 60
2, Portrait of a 26yr white woman, hyper-detailed, extremely ashamed, soft skin, 43                    Pick-a-Pic
3, zentai, 64
4, A close-up photograph of a fat orange cat with lasagna in its mouth. Shot on Leica M6., 63
5, toilet design toilet in style of dodge charger toilet, black, photo, 97
6, an attractive young woman rolling her eyes, 31
7, scarlett johansson, 56
8, RAW photo, aristocratic russian noblewoman, dressed in medieval dress, model face, come hither gesture, medieval mansion, medieval
9, Movie Still of The Joker wielding a red Lightsaber, Darth Joker a sinister evil clown prince of crime, HD Photograph, 21
10, The official portrait of an authoritarian president of an alternate america in 1960, in the style of pan am advertisements, looki
11, Beautiful young girl in the pool, 19
12, a closeup portrait of a playful maid, undercut hair, apron, amazing body, pronounced feminine feature, kitchen, , freckles, flirt
13, an cherry-colored dream, a fantasy painting, 33
14, Sign that says DON'T FART, 79
15, milim, pink hair, that awesome time i got reincarnated as a slime, 13
16, Orisa, 3
17, full glass of water standing on a mountain, 6
18, Card Magic the gathering style of tom whalen022 2e SM Ricardo. Lavage du char au gazole a Biesheim., 69
19, swirling water tornados epic fantasy, 68
```

```
0, Dwayne the Rock Johnson wrestles Jesus Christ in a WWE match in a hell in a cell., 8
1, An anime man in flight uniform with hyper detailed digital artwork and an art style inspired by Klimt, 37
2, A Wojak looking over a sea of memes from a cliff on 4chan., 1                                        HPD v2
3, Ralsei and Asriel from Deltarune eating pizza., 27
4, A portrait of an anime mecha robot with a Japanese town background and a starred night sky., 69
5, A minimalist portrait of Chloe Grace by Jean Giraud in a comic style., 60
6, A raccoon riding an oversized fox through a forest in a furry art anime still., 35
7, Chucky doll dressed as Beetlejuice., 72
8, 2B from NieR Automata eating a bagel., 47
9, Groot depicted as a flower., 81
10, A portrait of two women with purple hair flying in different directions against a dark background., 59
11, A girl with pink pigtails and face tattoos., 16
12, A cat in a tutu dancing to Swan Lake., 18
13, Wicked witch casting fireball dressed in green with screaming expression., 9
14, Link fights an octorok in a cave in a Don Bluth-style from The Legend of Zelda, 30
15, A cat with two horns on its head., 38
16, A cute anime schoolgirl with a sad face submerged in dark pink and blue water, 74
17, The image is a portrait of Homer Simpson as a Na'vi from Avatar, 12
18, A depiction of Chucky, 65
```

```
60, A collection of nail is sitting on a table., 3
61, A single clock is sitting on a table., 30
62, A couple of glasses are sitting on a table., 43                                                     DrawBench
63, An illustration of a large red elephant sitting on a small blue mouse., 62
64, An illustration of a small green elephant standing behind a large red mouse., 12
65, A small blue book sitting on a large red book., 28
66, A stack of 3 plates. A blue plate is on the top, sitting on a blue plate. The blue plate is in the middle, sitting on a green pla
67, A stack of 3 cubes. A red cube is on the top, sitting on a red cube. The red cube is in the middle, sitting on a green cube. The
68, A stack of 3 books. A green book is on the top, sitting on a red book. The red book is in the middle, sitting on a blue book. The
69, An emoji of a baby panda wearing a red hat, green gloves, red shirt, and green pants., 32
70, An emoji of a baby panda wearing a red hat, blue gloves, green shirt, and blue pants., 58
71, A fisheye lens view of a turtle sitting in a forest., 75
72, A side view of an owl sitting in a field., 83
73, A cross-section view of a brain., 2
74, A vehicle composed of two wheels held in a frame one behind the other, propelled by pedals and steered with handlebars attached t
75, A large motor vehicle carrying passengers by road, typically one serving the public on a fixed route and for a fare., 86
76, A small vessel propelled on water by oars, sails, or an engine., 56
77, A connection point by which firefighters can tap into a water supply., 78
78, A machine next to a parking space in a street, into which the driver puts money so as to be authorized to park the vehicle for a
79, A device consisting of a circular canopy of cloth on a folding metal frame supported by a central rod, used as protection against
```

Figure 18: Part of our test datasets.

