# OpenReview forum: "Noise Prompt Learning: Learning the Winning Tickets for Diffusion Sampling"
_ICLR.cc/2025/Conference — ICLR 2025 Conference Withdrawn Submission_

### Official Review · Reviewer_GoBd · 2024-10-29

**Soundness:** 3
**Presentation:** 3
**Contribution:** 2
**Rating:** 5
**Confidence:** 4

**Summary:**

This paper introduces the concept of a "noise prompt" to find optimal noise that enhances text-image alignment in generated images. The proposed method involves: (1) creating an NPD dataset with {Noise, Perturbed Noise} pairs containing high-quality noise, and (2) training NPNet, a noise-to-noise model. Comprehensive analysis demonstrates this approach generalizes across models and significantly improves text-image alignment. However, concerns arise regarding FID scores and comparisons with other techniques.

**Strengths:**

1. The paper demonstrates enhanced text-image alignment across various models, including SDXL, DreamShaper-xl-v2-turbo, and Hunyuan-DIT, substantiating its effectiveness in improving alignment.
2. It presents orthogonal experimentation by measuring performance when used alongside DPO. Given numerous works on improving text-image alignment, it raises the concern that performance gains may stem from fine-tuning with high-quality samples (in the perspective of RLHF) rather than NPNet’s inherent ability. The inclusion of DPO experiments indicates that noise manipulation may offer further improvement.
3. Introducing NPNet as a model for generating quality noise is novel.

**Weaknesses:**

1. Provide diversity metrics (e.g., FID, precision, recall [1]) across all models (SDXL, DreamShaper-xl-v2-turbo, Hunyuan-DIT). Currently, only FID on class-conditional ImageNet is reported, which seems limited. Bias toward certain noise could lower diversity, yet ImageNet’s small number of classes may skew results. Robust experiments showing that foundation text-to-image models maintain diversity would strengthen this claim. Although an aesthetic score is provided, it reflects only individual image quality.
2. Related work is in the Appendix and experimental results include only two comparative methods:
- 2-1. Move related work to the main paper. Repetition of text-image alignment results occupies substantial space. I recommend moving those to the Appendix and including related work in the main paper.
- 2-2. The compared technique Repaint is outdated. Use more recent methods.
- 2-3. Compare with Attend-and-Excite [2]. Though the Appendix states longer runtimes make it impractical, the paper specifies only a slight increase (5.6 to 9.7 seconds on Stable Diffusion), which is manageable.
- 2-4. The credibility of Appendix results for InitNO is questionable. Though the paper InitNO reported improved text-image alignment on StableDiffusion-v1-4, Table 6 shows InitNO’s alignment is lower than that of StableDiffusion-v1-4.
- 2-5. Details on Repaint and Inversion used in the main paper’s experiments are missing. Include these in the experiment section.
3. The illustration for Stage 2 in Figure 2 is unclear. According to the method section, noise calculated by the singular value predictor and the residual predictor is combined, and this combined noise is compared with inversed noise in loss calculation. However, the figure seems to imply separate loss measurements, which hinders clarity.

The two critical issues are: (1) whether the proposed method compromises diversity and (2) whether it performs better than other methods. I understand that InitNO or Attend-and-Excite can achieve higher performance due to its high computation time, therefore I would increase my score if the trade-off is acceptable. However, diversity is an important metric that should not be overlooked.

[1] Kynkäänniemi et al., "Improved Precision and Recall Metric for Assessing Generative Models."

[2]  Chefer et al., "Attend-and-Excite: Attention-Based Semantic Guidance for Text-to-Image Diffusion Model."

**Questions:**

1. Could you elaborate on the integration of DPO and NPNet? Was Stable Diffusion model fine-tuned by DPO first and later NPNet trained?

---

### Official Review · Reviewer_1619 · 2024-11-03

**Soundness:** 3
**Presentation:** 2
**Contribution:** 2
**Rating:** 5
**Confidence:** 4

**Summary:**

This work focuses on the problem of finding the best noise for a given text-condition used to generate images in the text-to-image diffusion models. This new noise is termed as noise prompt. It learns NPNet, a neural network which takes in input as a random noise and a text-prompt, and produces a new noise aka noise prompt. The new noise is a small perturbation on the input noise, which generates images aligning well with the given text-prompt.

To learn this NPNet, this work provides a dataset collection strategy to collect paired data. It collects the triplets $(x_T, x^{'}_T, c)$: input noise $x_T$, text-prompt $c$, and the target noise prompt $x^{'}_T$. Given a noise $x_T$ and text-prompt $c$, it uses DDIM denoising to get the denoised image at time step $T-1$. It then feeds in this image to DDIM-inversion procedure which adds noise to this input, and treats it as the target noise $x^{'}_T$. To get a higher quality dataset, it generates images using both the input and target noises, and if the image generated by the target noise is better than the input, it stores this triplet in the new dataset collection. This work uses a human preference reward model to judge the quality of the generated images.

Finally, it learns the NPNet, by optimizing the loss function on the collect noise prompt dataset. This entire framework is referred to as the Noise Prompt Learning framework.

**Strengths:**

- Problem formulation to find the optimal noise (aka noise prompt) given input noise and text-prompt seems interesting.
- Noise prompt dataset collection looks promising from the viewpoint of finding best noise given an input text-prompt and noise.
- NPNet seems to be applicable to various T2I models and produces good qualitative results with marginal inference overhead.

**Weaknesses:**

- Design of the NPNet architecture seems overly complex and its unclear why a small DiT/UNet style network (which takes noise and text-prompt as input and output target noise) would not suffice for this task? It seems relying on SVD decomposition of the input noise and predicting a residual error, might be too restrictive for a noise learning framework.
- Even the choice of loss function for training NPNet seems a bit unnatural. No ablations have been shown which gives any evidence that the vanilla MSE loss between predicted and target noise does not work.
- Lack of comparison against methods that perform noise optimization. Even though the paper mentions many existing works that perform noise optimization for image generation, it does not provide quantitative metrics against these methods.
- Quantitative improvements shown in the Table 1 and 2 look marginal compared to inversion baseline.

**Questions:**

- How to adapt this Noise Prompt learning framework to diffusion processes like rectified flow / EDM, etc.?
- For DDIM, instead of the above generation strategy, why not just sample two noises randomly instead of relying on DDIM-inversion?
- Why is the design for NPNet so complex? Wouldn't a simple network that takes in two inputs $x_T, c$ as conditions and output $x^{^}_T$ suffice? For instance a smaller variant of the original diffusion model?
- Do you have any ablations on the NPNet architecture?
- For training NPNet using the collected dataset, why doesn't simple MSE loss between predicted target noise and ground truth noise work?
- In many instances, results in quantitative metrics look very marginal compared to the baselines? How do you explain this behavior?
- Lack of comparisons with existing works that optimize noise for better quality generations in diffusion models. Only InitNO is compared in the appendix, have you done any comparison against other methods described in the related works (line 048-052).

---

### Official Review · Reviewer_eri4 · 2024-11-04

**Soundness:** 2
**Presentation:** 2
**Contribution:** 2
**Rating:** 5
**Confidence:** 3

**Summary:**

Given previous observations that certain noises serve as "winning tickets" for text-to-image diffusion models, to leverage this finding, the authors propose to train a lightweight network, NPNet, which transforms random Gaussian noise into these winning ticket noises tailored for specific text prompts. They compiled a dataset of 100k pairs of random and winning noises to facilitate this training. Experiments conducted across three diffusion models demonstrate that NPNet, as a plug-and-play module, enhances various image quality metrics, including PickScore, HPSv2, AES, ImageReward, CLIPScore, MPS, and FID.

**Strengths:**

1. Dataset Contribution: The creation and compilation of a 100k pair dataset of random and winning noises is a significant contribution to the field. This dataset can serve as a valuable resource for future research and experimentation.
2. Modular Approach: NPNet's design as a small, plug-and-play module ensures that it can be easily integrated into existing diffusion models without substantial overhead.
3. Comprehensive Evaluation: The authors evaluate NPNet across multiple diffusion models and a wide range of image quality metrics, providing a thorough assessment of its effectiveness.
4. Innovation in Noise Optimization: Introducing the concept of transforming Gaussian noise into winning tickets for specific prompts is a novel approach that could inspire further research in noise optimization for generative models.

**Weaknesses:**

1. Effectiveness Concerns: Despite the promising idea, the results presented in Fig. 1 and Table 1 indicate that the improvement in image quality metrics is marginal. This raises questions about the practical significance of NPNet's enhancements.
2. Some conclusions drawn from experiment results don’t look very convincing and consistent to me. For example, according to table 3, it is very hard to argue that NPNet is orthogonal to DPO, since only one metric (ImageReward) supports this argument, while the other three metrics don’t.

**Questions:**

1. Additional Baseline Comparisons: Incorporate additional baselines to strengthen the evaluation. For instance, for each text prompt, randomly sample multiple Gaussian noises (e.g., 32, 64, 128), process them through the diffusion model, and evaluate the image quality metrics. Comparing NPNet's performance against the best-performing random noise samples would provide a more robust benchmark. This can also serve as another method for dataset (pairs of random noise and corresponding winning tickets) construction.
2. Dataset Accessibility: Do the authors plan to release the collected dataset publicly? As mentioned in the strengths, releasing the collected dataset would greatly benefit the research community. This transparency would allow others to validate results and build upon the work.

---

### Official Review · Reviewer_QDRm · 2024-11-04

**Soundness:** 3
**Presentation:** 2
**Contribution:** 3
**Rating:** 3
**Confidence:** 4

**Summary:**

This paper proposes a novel framework to achieve winning tickets in text-to-image diffusion generation. It identifies a new concept of noise prompt, turning a random initial noise into an effective noise for text alignment, and formulates the noise prompt learning framework. Also, it collects a new noise prompt dataset (NPD) to train a noise prompt network (NPNet) that generates winning noise tickets. Finally, it demonstrates the generalization and effectiveness of the framework through extensive experiments.

**Strengths:**

- This paper proposes a novel concept of noise prompt learning, an efficient machine-learning based framework to refine Gaussian noise.
- The way this paper tackles the problem is interesting and intuitive.
- Extensive experiments to verify the effectiveness of the proposed method are presented.

**Weaknesses:**

- While the design choice of NPNet is one of the most important contributions of this paper, the detailed explanation and rationales are lacking except for Fig 5.
- The number of test prompts for test looks relatively small (100, 100, and 200 each) and the performance improvements in table 1 and 2 are negligible to demonstrate the superiority of the proposed method.
- For better understanding of the effects of NPNet, in-depth analysis of the differences between source and predicted noise should be presented. Do the differences in noise space have some semantic relationships to text prompts? Moreover, how does NPNet improve the compositionality of diffusion models as shown in Figure 1?

**Questions:**

- How is CFG scale used in DDIM-inversion?
- What is the reason for not training NPNet solely for Hunyuan-DiT?

---

### Official Review · Reviewer_d6fL · 2024-11-05

**Soundness:** 3
**Presentation:** 3
**Contribution:** 2
**Rating:** 6
**Confidence:** 4

**Summary:**

This paper introduces noise prompt learning, a framework for learning noise 'correction' networks that output winning-ticket noise conditioned on text prompt, accompanying a dataset for learning such networks.
The authors treated the input noise to diffusion models as part of the prompts, advocating that using a network to refine such prompts can improve the generated image quality. To train such networks, a dataset is collected where iDDIM is used to generate 'better' noise at $x_T$ as positive samples and the original noise as the negative samples, with a preference model to facilitate the quality filtering. With such a dataset, a noise refinement network can be trained to predict the correction to an input noise given text conditions. The effectiveness of the proposed method is evaluated on multiple models and datasets/

**Strengths:**

- The proposed method is very intuitive, and the presentation, in general, is very easy to follow. The figures help explain the proposed method and data collection pipeline well.

- The analysis of the singular values of the noise is very interesting, and the subsequent noise prediction targets seem very useful.

**Weaknesses:**

- Positioning of this method

My biggest concern about this paper is how to position this proposed method properly. As indicated by the authors, the winning-ticket noise obtained through iDDIM contains more semantics than the standard noise. So intuitively, another way of understanding this method is that the NPNet is actually a very small-scale human-preference distilled version of the diffusion model, focusing exclusively on the very first or first few denoising steps.

- Quantitative results

The aforementioned concern is further amplified by the tables, where in most settings and metrics, the proposed method only improves over baselines and related methods marginally.

**Questions:**

- Considering the semantics reside in the winning-ticket noise, after obtaining them, do we really have to go through all denoising steps again? How would the performance change if we directly use the NPNet output as, for example, $x_{T-1}$?

- How will the NPNet perform without any further finetuning when transferring to a new model?

- I find a preprint work [1] that might be relevant to the proposed method.

[1] Model-Agnostic Human Preference Inversion in Diffusion Models, ArXiv 2024.

---

### Author Response · Authors · 2024-11-14

Dear Reviewers,

We thank you for your dedication and valuable suggestions. After careful consideration, we have decided to withdraw the manuscript. In future versions, we will refine both the presentation and address the methodological and experimental deficiencies.

Best Wishes,

Authors

---

### Note · Authors · 2024-11-14

I have read and agree with the venue's withdrawal policy on behalf of myself and my co-authors.